# Preserved sensory processing but hampered conflict detection when stimulus input is task-irrelevant

Stijn Adriaan Nuiten[1,2]*, Andrés Canales-Johnson[1,2,3,4], Lola Beerendonk[1,2,3], Nutsa Nanuashvili[1,2], Johannes Jacobus Fahrenfort[1,2], Tristan Bekinschtein[3,5†], Simon van Gaal[1,2†]*

[1]Department of Psychology, University of Amsterdam, Amsterdam, Netherlands; [2]Amsterdam Brain & Cognition, University of Amsterdam, Amsterdam, Netherlands; [3]Department of Psychology, University of Cambridge, Cambridge, United Kingdom; [4]Vicerrectoría de Investigación y Posgrado, Universidad Católica del Maule, Talca, Chile; [5]Behavioural and Clinical Neuroscience Institute, Cambridge, United Kingdom

**Abstract** Conflict detection in sensory input is central to adaptive human behavior. Perhaps unsurprisingly, past research has shown that conflict may even be detected in the absence of conflict awareness, suggesting that conflict detection is an automatic process that does not require attention. To test the possibility of conflict processing in the absence of attention, we manipulated task relevance and response overlap of potentially conflicting stimulus features across six behavioral tasks. Multivariate analyses on human electroencephalographic data revealed neural signatures of conflict only when at least one feature of a conflicting stimulus was attended, regardless of whether that feature was part of the conflict, or overlaps with the response. In contrast, neural signatures of basic sensory processes were present even when a stimulus was completely unattended. These data reveal an attentional bottleneck at the level of objects, suggesting that object-based attention is a prerequisite for cognitive control operations involved in conflict detection.

*For correspondence:
stijnnuiten@gmail.com (SAN);
simonvangaal@gmail.com (SvG)

†These authors contributed equally to this work

Competing interests: The authors declare that no competing interests exist.

## Introduction

Every day we are bombarded with sensory information from the environment, and we often face the challenge of selecting the relevant information and ignoring irrelevant – potentially conflicting – information to maximize performance. These selection processes require much effort and our full attention, sometimes rendering us deceptively oblivious to irrelevant sensory input (e.g., chest-banging apes), as illustrated by the famous inattentional blindness phenomenon (*Simons and Chabris, 1999*). However, unattended events that are not relevant for the current task might still capture our attention or interfere with ongoing task performance, for example, when they are inherently relevant to us (e.g., our own name). This is illustrated by another famous psychological phenomenon: the cocktail party effect (*Cherry, 1953*; *Moray, 1959*). Thus, under specific circumstances, task-irrelevant information may capture attentional resources and be subsequently processed with different degrees of depth.

It is currently a matter of debate which processes require top-down attention (*Dehaene et al., 2006*; *Koch and Tsuchiya, 2007*; *Koelewijn et al., 2010*; *Lamme, 2003*; *Lamme and Roelfsema, 2000*; *Rousselet et al., 2004*; *VanRullen, 2007*). It was long thought that only basic physical stimulus features or very salient stimuli are processed in the absence of attention (*Treisman and Gelade, 1980*) due to an 'attentional bottleneck' at higher levels of analysis (*Broadbent, 1958*; *Deutsch and Deutsch, 1963*; *Lachter et al., 2004*; *Wolfe and Horowitz, 2004*). However, there is now solid

**eLife digest :** Focusing your attention on one thing can leave you surprisingly unaware of what goes on around you. A classic experiment known as 'the invisible gorilla' highlights this phenomenon. Volunteers were asked to watch a clip featuring basketball players, and count how often those wearing white shirts passed the ball: around half of participants failed to spot that someone wearing a gorilla costume wandered into the game and spent nine seconds on screen.

Yet, things that you are not focusing on can sometimes grab your attention anyway. Take for example, the 'cocktail party effect', the ability to hear your name among the murmur of a crowded room. So why can we react to our own names, but fail to spot the gorilla? To help answer this question, Nuiten et al. examined how paying attention affects the way the brain processes input.

Healthy volunteers were asked to perform various tasks while the words 'left' or 'right' played through speakers. The content of the word was sometimes consistent with its location ('left' being played on the left speaker), and sometimes opposite ('left' being played on the right speaker). Processing either the content or the location of the word is relatively simple for the brain; however detecting a discrepancy between these two properties is challenging, requiring the information to be processed in a brain region that monitors conflict in sensory input.

To manipulate whether the volunteers needed to pay attention to the words, Nuiten et al. made their content or location either relevant or irrelevant for a task. By analyzing brain activity and task performance, they were able to study the effects of attention on how the word properties were processed.

The results showed that the volunteers' brains were capable of dealing with basic information, such as location or content, even when their attention was directed elsewhere. But discrepancies between content and location could only be detected when the volunteers were focusing on the words, or when their content or location was directly relevant to the task.

The findings by Nuiten et al. suggest that while performing a difficult task, our brains continue to react to basic input but often fail to process more complex information. This, in turn, has implications for a range of human activities such as driving. New technology could potentially help to counteract this phenomenon, aiming to direct attention towards complex information that might otherwise be missed.

evidence that several tasks may in fact still unfold in the (near) absence of attention, including perceptual integration (*Fahrenfort et al., 2017*), the processing of emotional valence (*Sand and Wiens, 2011*; *Stefanics et al., 2012*), semantic processing of written words (*Schnuerch et al., 2016*), and visual scene categorization (*Li et al., 2002*; *Peelen et al., 2009*). Although one should be cautious in claiming complete absence of attention (*Lachter et al., 2004*), these and other studies have pushed the boundaries of input processing that is task-irrelevant (without attention) and may even question the existence of an attentional bottleneck at all, at least for relatively low-level information. Conceivably, the attentional bottleneck is only present at higher, more complex, levels of cognitive processing, like cognitive control functions.

Over the years, various theories have been proposed with regard to this attentional bottleneck among which are the load theory of selective attention and cognitive control (*Lavie et al., 2004*), the multiple resources theory (*Wickens, 2002*), and the hierarchical central executive bottleneck theory and formalizations thereof in a cortical network model for serial and parallel processing (*Sigman and Dehaene, 2006*; *Zylberberg et al., 2010*; *Zylberberg et al., 2011*). These theories all hinge on the idea that resources for the processing of information are limited and that the brain therefore has to allocate resources to processes that are currently most relevant via selective attention (*Broadbent, 1958*; *Treisman, 1969*). Resource (re-)allocation, and thus flexible behavior, is thought to be governed by an executive network, most prominently involving the prefrontal cortex (*Goldman-Rakic, 1995*; *Goldman-Rakic, 1996*). Information that is deemed task-irrelevant has fewer resources at its disposal and is therefore processed to a lesser extent. When more resources are necessary for processing the task-relevant information, for example, under high perceptual load, processing of task-irrelevant information diminishes (*Lavie et al., 2003*; *Lavie et al., 2004*). Yet even under high perceptual load, task-irrelevant features can be processed when they are part of an

attended object (when object-based attention is present) (*Chen, 2012*; *Chen and Cave, 2006*; *Cosman and Vecera, 2012*; *Kahneman et al., 1992*; *O'Craven et al., 1999*; *Schoenfeld et al., 2014*; *Wegener et al., 2014*). There is currently no consensus which type of information can be processed in parallel by the brain and which attentional mechanisms determine what information passes the attentional bottleneck. One unresolved issue is that most empirical work has investigated the bottleneck with regard to sensory features; however, it is unknown if the bottleneck and the distribution of processing resources also take place for more complex, cognitive processes. Here, we test whether such a high-level attentional bottleneck indeed exists in the human brain.

Specifically, we aim to test whether cognitive control operations, necessary to identify and resolve conflicting sensory input, are operational when that input is irrelevant for the task at hand (and hence unattended) and what role object-based attention may have in conflict detection. Previous work has shown that the brain has dedicated networks for the detection and resolution of conflict, in which the medial frontal cortex (MFC) plays a pivotal role (*Ridderinkhof et al., 2004*). Conflict detection and subsequent behavioral adaptation is central to human cognitive control, and, hence, it may not be surprising that past research has shown that conflict detection can even occur unconsciously (*Atas et al., 2016*; *D'Ostilio and Garraux, 2012a*; *Huber-Huber and Ansorge, 2018*; *van Gaal et al., 2008*), suggesting that the brain may detect conflict fully automatically and that it may even occur without paying attention (e.g., *Rahnev et al., 2012*). Moreover, it has been shown that this automaticity can be enhanced by training, resulting in more efficient processing of conflict (*Chen et al., 2013*; *MacLeod and Dunbar, 1988*; *van Gaal et al., 2008*).

Conclusive evidence regarding the claim that conflict detection is fully automatic has, to our knowledge, not been provided, and therefore, the necessity of attention for cognitive control operations remains open for debate. Previous studies have shown that cognitive control processes are operational when to-be-ignored features from either a task-relevant or a task-irrelevant stimulus overlap with the behavioral response to be made to the primary task, causing interference in performance (*Mao and Wang, 2008*; *Padrão et al., 2015*; *Zimmer et al., 2010*). In these circumstances, the interfering stimulus feature carries information related to the primary task and is therefore *de facto* not task-irrelevant. Consequently, it is currently unknown whether cognitive control operations are active for conflicting sensory input that is not related to the task at hand. Given the immense stream of sensory input we encounter in our daily lives, conflict between two (unattended) sources of perceptual information is inevitable.

Here, we investigated whether conflict between two features of an auditory stimulus (its content and its spatial location) would be detected by the brain under varying levels of task relevance of these features. The main aspect of the task was as follows. We presented auditory spoken words ('left' and 'right' in Dutch) through speakers located on the left and right side of the body. By presenting these stimuli through either the left or the right speaker, content-location conflict arises on specific trials (e.g., the word 'left' from the right speaker) but not on others (e.g., the word 'right' from the right speaker) (*Buzzell et al., 2013*; *Canales-Johnson et al., 2020*). A wealth of previous studies has revealed that conflict arises between task-relevant and task-irrelevant features of the stimulus in these type of tasks (similar to the Simon task and Stroop task; *Egner and Hirsch, 2005*; *Hommel, 2011*). Here, these potentially conflicting auditory stimuli were presented during six different behavioral tasks, divided over two separate experiments, multiple experimental sessions, and different participant groups (both experiments N = 24). In all tasks, we focus on the processing of content-location conflict of the auditory stimulus. There were several critical differences between the behavioral tasks: (1) task relevance of a conflicting feature of the stimulus, (2) task relevance of a non-conflicting feature that was part of a conflicting stimulus, and (3) whether the response to be given mapped onto a conflicting feature of the stimulus. Note that in all tasks only one feature could be task-relevant and that all the other feature(s) had to be ignored. The systematic manipulation of task relevance and the response-mapping allowed us to explore the full landscape of possibilities of how varying levels of attention affect sensory and conflict processing. Electroencephalography (EEG) was recorded and multivariate analyses on the EEG data were used to extract any neural signatures of conflict detection (i.e., theta-band neural oscillations; *Cavanagh and Frank, 2014*; *Cohen and Cavanagh, 2011*) and sensory processing for any of the features of the auditory stimulus. Furthermore, in both experiments we measured behavioral and neural effects of task-irrelevant conflict before and after training on conflict-inducing tasks, aiming to investigate the role of automaticity in the detection of (task-irrelevant) conflict.

## Results

### Experiment 1: can the brain detect fully task-irrelevant conflict?

In the first experiment, 24 human participants performed two behavioral tasks (*Figure 1A*). In the auditory conflict task (from hereon: content discrimination task I), the feature 'sound content' was task-relevant. Participants were instructed to respond according to the content of the auditory stimulus ('left' vs. 'right'), ignoring its spatial location that could conflict with the content response (presented from the left or right side of the participant). For the other behavioral task, participants performed a demanding visual random dot-motion (RDM) task in which they had to discriminate the direction of vertical motion (from hereon: vertical RDM task), while being presented with the same auditory stimuli – all features of which were thus fully irrelevant for task performance. Behavioral responses on this visual task were orthogonal to the response tendencies potentially triggered by the auditory features, excluding any task- or response-related interference (*Figure 1B*). Under this manipulation, all auditory features are task-irrelevant and are orthogonal to the response-mapping. To maximize the possibility of observing conflict detection when conflicting features are task-irrelevant and explore the effect of task automatization on conflict processing, participants performed the tasks both before and after extensive training, which may increase the efficiency of cognitive control (*Figure 1C*; *van Gaal et al., 2008*).

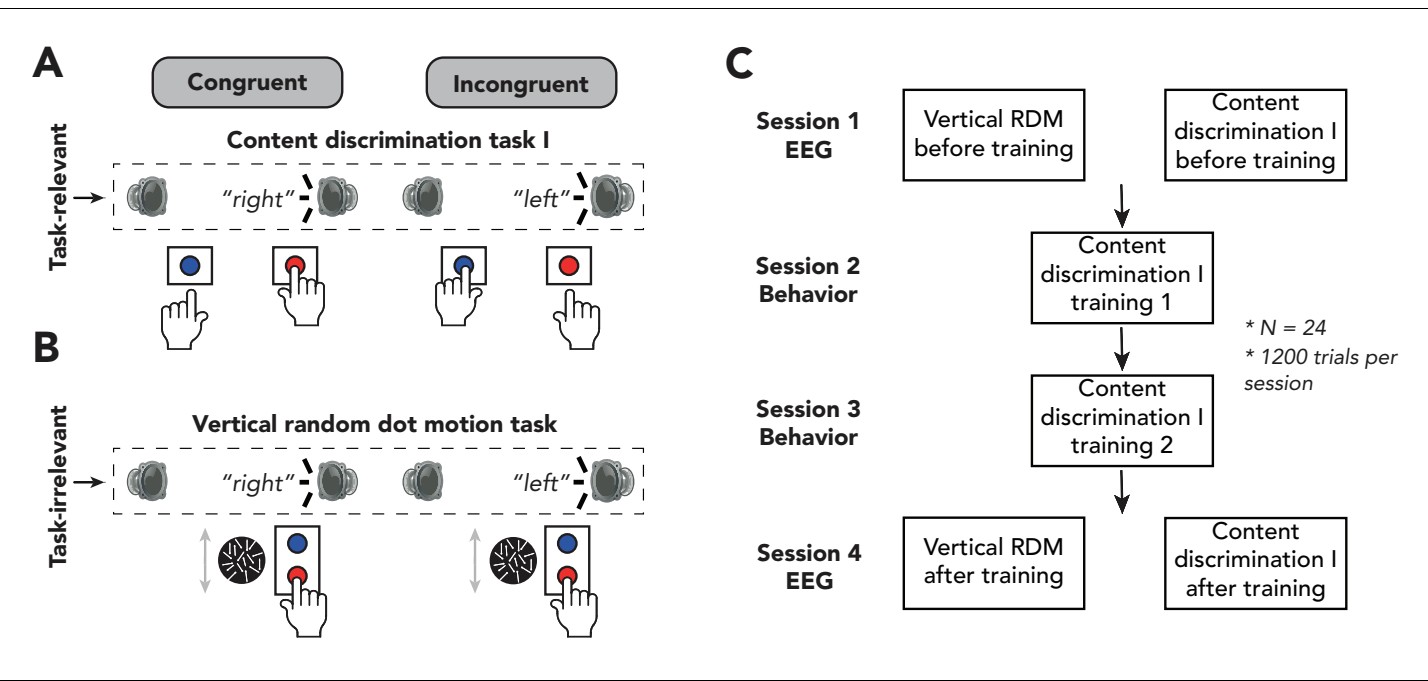

**Figure 1.** Experimental design of experiment 1. (**A, B**) Schematic representation of the experimental design for auditory content discrimination task I (**A**) and vertical random dot-motion (RDM) task (**B**). In both tasks, the spoken words 'left' and "right were presented through either a speaker located on the left or right side of the participant. Note that auditory stimuli are only task-relevant in auditory content discrimination task I and not in the vertical RDM task. In this figure, sounds are only depicted as originating from the right, whereas in the experiment the sounds could also originate from the left speaker. (**A**) In content discrimination task I, participants were instructed to report the content ('left' or 'right') of an auditory stimulus via a button press with their left or right hand, respectively, and to ignore the spatial location at which the auditory stimulus was presented. (**B**) During the vertical RDM task, participants were instructed to report the overall movement direction of dots (up or down) via a button press with their right hand, whilst still being presented with the auditory stimuli, which were therefore task-irrelevant. In both tasks, content of the auditory stimuli could be congruent or incongruent with its location of presentation (50% congruent/incongruent trials). (**C**) Overview of the sequence of the four experimental sessions of this study. Participants performed two electroencephalography sessions during which they first performed the vertical RDM task followed by auditory content discrimination task I. Each session consisted of 1200 trials, divided over 12 blocks, allowing participants to rest in between blocks. In between experimental sessions, participants were trained on auditory content discrimination task I on two training sessions of 1 hr each.

## Experiment 1: conflicting information induces slower responses and decreased accuracy only for task-relevant sensory input

For content discrimination task I, mean error rates (ERs) were 2.6% ($SD$ = 2.7%) and mean reaction times (RTs) 477.2 ms ($SD$ = 76.1 ms), averaged over all four sessions. For the vertical RDM, mean ERs were 19.2% ($SD$ = 6.6%) and mean RTs were 711.4 ms ($SD$ = 151.3 ms). The mean ER of vertical RDM indicates that our staircasing procedure was effective (see Materials and methods for details on staircasing performance on the RDM). To investigate whether our experimental design was apt to induce conflict effects for task-relevant sensory input and to test whether conflict effects were still present when sensory input was task-irrelevant, we performed repeated measures (rm-)ANOVAs (2 × 2 × 2 factorial) on mean RTs and ERs gathered during the EEG recording sessions (session 1, 'before training'; session 4, 'after training'). This allowed us to include (1) task relevance (yes/no), (2) training (before/after), and (3) congruency of auditory content with location of auditory source (congruent/incongruent). Note that congruency is always defined based on the relationship between two features of the auditorily presented stimuli, also when participants performed the visual task (and therefore the auditory features were task-irrelevant).

Detection of conflict is typically associated with behavioral slowing and increased ERs. Indeed, we observed that, across both tasks, participants were slower and made more errors on incongruent trials as compared to congruent trials (the conflict effect, RT: $F(1,23)$ = 52.83, p<0.001, $\eta_p^2$ = 0.70; ER: $F(1,23)$ = 9.13, p=0.01, $\eta_p^2$ = 0.28). This conflict effect was modulated by task relevance of the auditory features (RT: $F(1,23)$ = 152.76, p<0.001, $\eta_p^2$ = 0.87; ER: $F(1,23)$ = 11.15, p=0.01, $\eta_p^2$ = 0.33) and post-hoc ANOVAs (see Materials and methods) showed that the conflict effect was present when the auditory feature content was task-relevant ($RT_{cont(I)}$: $F(1,23)$ = 285.00, p<0.001, $\eta_p^2$ = 0.93; $ER_{cont(I)}$: $F(1,23)$ = 23.85, p<0.001, $\eta_p^2$ = 0.51; *Figure 2A*, left panel), but not when all auditory features were task-irrelevant ($RT_{VRDM}$: $F(1,23)$ = 1.96, p=0.18, $\eta_p^2$ = 0.08, $BF_{01}$ = 5.41; $ER_{VRDM}$: $F(1,23)$ = 0.26, p=0.62, $\eta_p^2$ = 0.01, $BF_{01}$ = 4.55; *Figure 2A*, right panel). Because responses in the vertical RDM were made with the right hand only, we subsequently tested whether the auditory features *in isolation* affected the speed and accuracy of right-hand responses. For example, the spoken word 'left' may slow down responses made with the right hand more so than the spoken word 'right' (the same logic holds for stimulus location). However, this was not the case. A 2 × 2 × 2 factorial rm-ANOVA on mean RTs with session (before/after training), stimulus content ('left'/'right'), and stimulus location (left/right) showed that RTs were unaffected by sound content ($F(1,23)$ = 0.01, p=0.92, $\eta_p^2$ = 0.00, $BF_{01}$ = 6.16) and sound location ($F(1,23)$ = 0.49, p=0.49, $\eta_p^2$ = 0.02, $BF_{01}$ = 6.36).

Participants performed both behavioral tasks before and after extensive training of the content discrimination task to be able to investigate the role of training on conflict processing (*Figure 1C*). RTs and ERs in the vertical RDM task were not modulated by behavioral training ($RT_{VRDM}$: $F(1,23)$ = 2.07, p=0.16, $\eta_p^2$ = 0.08, $BF_{01}$ = 0.32; $ER_{VRDM}$: $F(1,23)$ = 0.24, p=0.63, $\eta_p^2$ = 0.01, $BF_{01}$ = 3.79). Training did result in a decrease of overall RT on content discrimination task I, although ERs were not affected ($RT_{cont(I)}$: $F(1,23)$ = 45.05, p<0.001, $\eta_p^2$ = 0.66; $ER_{cont(I)}$: $F(1,23)$ = 1.77, p=0.20, $\eta_p^2$ = 0.07, $BF_{01}$ = 0.89). Moreover, the effect of conflict on RTs and ERs in this task decreased after behavioral training ($RT_{cont(I)}$: $F(1,23)$ = 29.86, p<0.001, $\eta_p^2$ = 0.57; $ER_{cont(I)}$: $F(1,23)$ = 9.76, p=0.005, $\eta_p^2$ = 0.30; *Figure 2—figure supplement 1A, B*), suggesting increased efficiency of within-trial conflict resolution mechanisms. All other effects were not reliable (p>0.05).

## Experiment 1: neural signatures of conflict detection only for task-relevant stimuli

The observation that conflicting task-irrelevant stimuli had no effect on RTs and ERs, even after substantial training, whereas task-relevant conflicting stimuli did, may not come as a surprise because manual responses on the visual task (motion up/down with index and middle finger of right hand) were fully orthogonal to the potential conflicting nature of the auditory features (i.e., left/right). Further, content discrimination task I and the vertical RDM were independent tasks, requiring different cognitive processes. For example, mean RTs on the vertical RDM were on average 267 ms longer than mean RTs for content discrimination task I. However, caution is required in concluding that conflict detection is absent for task-irrelevant stimuli based on these behavioral results alone as neural

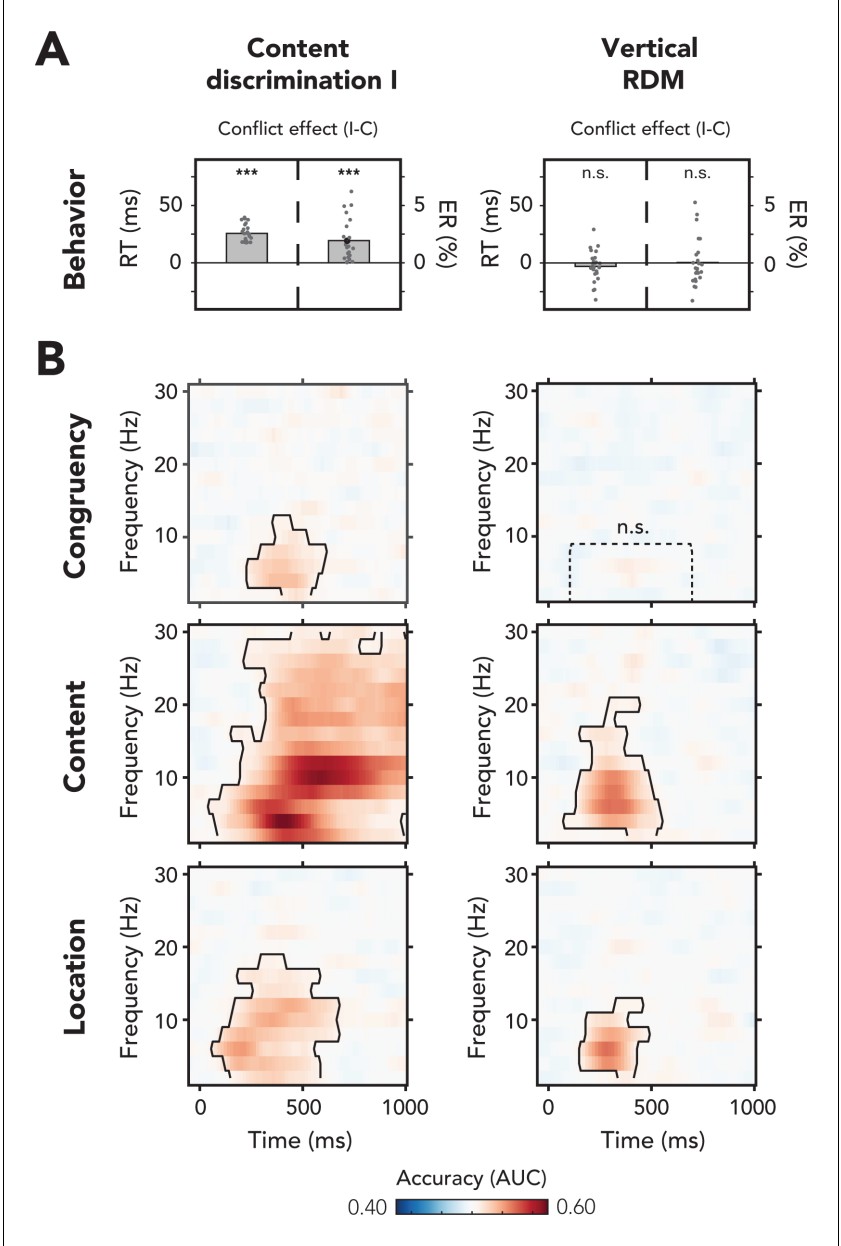

**Figure 2.** Behavioral and multivariate decoding results of experiment 1. (**A, B**) All results depicted here are from the merged data of both experimental sessions. The left column of plots shows the results for content discrimination task I, where auditory stimuli and conflicting features were task-relevant. The right column of plots shows the results for the vertical random dot-motion (RDM), where neither the auditory stimulus nor its conflicting features were task-relevant. (**A**) The behavioral results are plotted as conflict effects (incongruent – congruent). Effects of conflict were present in content discrimination task I, with longer reaction times (RTs) (left bar) and increased error rates (ERs) (right bar) for incongruent compared to congruent trials. For the vertical RDM task, no significant effects of conflict were found in behavior. Dots represent individual participants. The behavioral data that is shown here can be found in *Figure 2—source data 1*. (**B**) Multivariate classifier accuracies for different stimulus features. We trained classifiers on three stimulus features: auditory congruency, auditory content, and auditory location. Classifier accuracies (area under the curve [AUC]) are plotted across a time-frequency window of −100 ms to 1000 ms and 2–30 Hz. Classifier accuracies are thresholded (cluster-based corrected, one-sided: $\bar{X}>0.5$, $p<0.05$), and significant clusters are outlined with a solid black line. The dotted box shows the predefined ROI on which we performed a hypothesis-driven analysis. The classifier accuracies within this ROI were not significantly greater than chance for the vertical RDM task. Note that conflicting features of the auditory stimulus, content and location, could be decoded from neural data regardless of attention to the auditory stimulus.
*Figure 2 continued on next page*

*Figure 2 continued*

Information related to auditory congruency was present in a theta-band cluster, but only when the auditory stimulus was attended. *** p<0.001, n.s.: p>0.05.

The online version of this article includes the following source data and figure supplement(s) for figure 2:

**Source data 1.** Behavioral results of experiment 1.
**Figure supplement 1.** Effects of behavioral training on behavioral effects of conflict and decoding performance in experiment 1.
**Figure supplement 1—source data 1.** Behavioral results of experiment 1 - before and after training.

and/or behavioral effects can sometimes be observed in isolation (one is observed but not the other, e.g., *Canales-Johnson et al., 2020*; *van Gaal et al., 2014*). Therefore, in order to test whether unattended conflict is detected by the brain we turn to the multivariate pattern analysis (MVPA) of our neural data.

Plausibly, the neural dynamics of conflict processing for task-irrelevant sensory input are different – in physical (across electrodes) and frequency space – from those related to the processing of conflict when sensory input is task-relevant. Therefore, we applied multivariate decoding techniques in the frequency domain to inspect whether and – if so – to what extent certain stimulus features were processed. These multivariate approaches have some advantages over traditional univariate approaches, for example, they are less sensitive to individual differences in spatial topography, because decoding accuracies are derived at a single participant level (*Fahrenfort et al., 2018*; *Grootswagers et al., 2017*; *Haxby et al., 2001*). Therefore, group statistics do not critically depend on the presence of effects in specific electrodes or clusters of electrodes. Further, although a wealth of studies have shown that conflict processing is related to an increase in power of theta-band neural oscillations (~4–8 Hz) after stimulus presentation (*Cohen and Cavanagh, 2011*; *Jiang et al., 2015a*; *Nigbur et al., 2012*), it is unknown whether this is also the case for task-irrelevant conflict. By performing our MVPA in frequency space, we could potentially find neural signatures in non-theta frequency bands related to the processing of task-irrelevant conflict. However, due to the temporal and frequency space that has to be covered, strict multiple comparison corrections have to be performed (across time and frequency, see Materials and methods). Therefore, we adopted an additional hypothesis-driven analysis, which also allowed us to obtain evidence for the absence of effects. Throughout this paper, we will discuss our neural data in the following order. First, the MVPAs in the frequency domain are presented for all critical features of the task (congruency, content, location, corrected for multiple comparisons). Then, we report results from the additional hypothesis-driven analysis, where we extracted classifier accuracies from a predefined time-frequency region of interest (ROI) (100–700 ms, 2–8 Hz) on which we performed (Bayesian) tests (see Materials and methods). This ROI was selected based on previous observations of conflict-related theta-band activity (*Cohen and Cavanagh, 2011*; *Cohen and van Gaal, 2014*; *Jiang et al., 2015b*; *Nigbur et al., 2012*). Specifically, for every task and every stimulus feature (i.e., congruency, content, location), we extracted average decoding accuracies from the ROI per participant and performed analyses on these values.

First, we trained a classifier on data from all EEG electrodes to distinguish between congruent versus incongruent trials, for both content discrimination task I and the vertical RDM task. Above-chance classification accuracies imply that relevant information about the decoded stimulus feature is present in the neural data, meaning that some processing of that feature occurred (*Hebart and Baker, 2018*). We performed our main analysis on the combined data from both EEG sessions, thereby maximizing power to establish effects in our crucial comparisons. We also performed similar analyses on the session-specific data to investigate the role of behavioral training on processing of conflict. These results are discussed more in depth below and are shown in *Figure 2—figure supplement 1C, D*.

Congruency decoding reveals that stimulus congruency was represented in neural data only when conflict was task-relevant (p<0.001, one-sided: $\bar{X}$>0.5, cluster-corrected; frequency range: 2–12 Hz, peak frequency: 4 Hz, time range: 234–609 ms, peak time: 438 ms; *Figure 2B*, left panel). The conflict effect roughly falls in the theta-band (4–8 Hz), which confirms a firm body of literature linking conflict detection to post-conflict modulations in theta-band oscillatory dynamics (*Cavanagh and*

*Frank, 2014*; *Cohen and Cavanagh, 2011*; *Cohen and van Gaal, 2014*; *Nigbur et al., 2012*). Activation patterns that were calculated from classifier weights within the predefined time-frequency theta-band ROI (2–8 Hz, 100–700 ms) revealed a clear midfrontal distribution of conflict-related activity (*Figure 5—figure supplement 1A*). No significant time-frequency cluster was found for the vertical RDM task (*Figure 2B*, right panel). To quantify the absence of this effect, we followed up this hypothesis-free (with respect to frequency and time) MVPA with a hypothesis-driven analysis focused on the post-stimulus theta-band. This more restricted analysis showed no significant effect ($t$(23) = −0.50, p=0.69, $d$ = −0.10) and an additional Bayesian analysis revealed moderate evidence in favor of the null hypothesis (i.e., no effect of conflict on theta-band power) than the alternative hypothesis ($BF_{01}$ = 6.53).

Similar to our observation of decreased behavioral effects of conflict after behavioral training (*Figure 2—figure supplement 1A, B*), decoding accuracies in the content discrimination task I were also lower after training ($t$(23) = −3.01, p=0.01, $d$ = −0.63; *Figure 2—figure supplement 1C*), suggesting more efficient conflict resolution, as reflected in neural theta oscillations as well. In the vertical RDM, behavioral training did not affect decoding accuracies of sound congruency ($t$(23) = −1.24, p=0.23, $d$ = −0.25, $BF_{01}$ = 2.36; *Figure 2—figure supplement 1D*).

## Experiment 1: stimulus features are processed in parallel, independent of task relevance

Thus, cognitive control networks − or possible substitute networks − are seemingly not capable of detecting conflict when sensory features are task-irrelevant. However, the question remains whether this observation is specific to the conflicting nature of the auditory stimuli or whether the auditory stimuli are not processed whatsoever when attention is reallocated to the visually demanding task. To address this question, we trained classifiers on two other features of the auditory stimuli, that is, location and content, to test whether these features were processed by the brain regardless of task relevance. Indeed, the content of auditory stimuli was processed both when the stimuli were task-relevant (p<0.001, one-sided: $\bar{X}$>0.5, cluster-corrected; frequency range: 2–30 Hz, peak frequency: 4 Hz, time range: 47–1000 ms, peak time: 422 ms; *Figure 2B*, left panel) and task-irrelevant (p<0.001, one-sided: $\bar{X}$>0.5, cluster-corrected; frequency range: 2–20 Hz, peak frequency: 6 Hz, time range: 78–547 ms, peak time: 297 ms; *Figure 2B*, right panel).

Similarly, the location of auditory stimuli could also be decoded from neural data for both content discrimination task I (p<0.001, one-sided: $\bar{X}$>0.5, cluster-corrected; frequency range: 2–18 Hz, peak frequency: 6 Hz, time range: 63–-672 ms, peak time: 203 ms; *Figure 2B*, left panel) and the vertical RDM task (p<0.001, one-sided: $\bar{X}$>0.5, cluster-corrected; frequency range: 2–12 Hz, peak frequency: 6 Hz, time range: 156–484 ms, peak time: 281 ms; *Figure 2B*, right panel). The above chance performance of the classifiers for the auditory stimulus features demonstrates that location and content information were processed, even when these features were task-irrelevant. Processing of task-irrelevant stimulus features was, however, more transient in time and more narrowband in frequency as compared to processing of the same features in a task-relevant setting. Further, content decoding revealed a much broader frequency spectrum than any of the other comparisons in content discrimination task I. In the next experiment, we show that this is related to the fact that this feature was response-relevant and that this effect therefore partially reflects response preparation and response execution processes. Summarizing, we show that when (conflicting) features of an auditory stimulus are truly and consistently task-irrelevant, the conflict between them is no longer detected by – nor relevant to – the conflict monitoring system, but the features (content and location) are still processed in isolation.

To investigate if, and how, behavioral training affects processing of sound content and location, we tested whether decoding accuracies for these features were different between the two experimental sessions. Decoding accuracies for the task-relevant feature of content discrimination task I (i.e., sound content) were significantly increased after behavioral training in a delta- to theta-band cluster (p<0.001, one-sided: $\bar{X}$>0.5, cluster-corrected; frequency range: 2–10 Hz, peak frequency: 2 Hz, time range: 234–484 ms, peak time: 344 ms; *Figure 2—figure supplement 1C*). This suggests that processing of task-relevant information (i.e., sound content) is improved as a result of training. Decoding accuracies for sound location in the content discrimination task were not different before and after behavioral training (no significant clusters; predefined ROI: $t$(23) = 0.12, p=0.91, $d$ = 0.02,

$BF_{01}$ = 4.63). In the vertical RDM task, behavioral training did not affect the decoding accuracies within the predefined ROI for sound content ($t(23)$ = 0.75, p=0.46, $d$ = 0.15, $BF_{01}$ = 3.61) and location ($t(23)$ = 0.04, p=0.97, $d$ = 0.01, $BF_{01}$ = 4.66, also no other significant clusters; *Figure 2—figure supplement 1D*). This suggests that processing of sound content and location (*Figure 2B*), both task-irrelevant auditory features in the vertical RDM task, is automatic and not dependent on training.

In conclusion, we observed neural signatures of the processing of sensory stimulus features (i.e., location and content of an auditory stimulus) regardless of task relevance of these features, but a lack of integration of these features to form conflict when the auditory stimulus was fully task-irrelevant. Considerable training in content discrimination task I resulted in more efficient conflict processing (i.e., decreased behavioral conflict effects and theta-band activity after training; *Figure 2—figure supplement 1*) when the auditory stimulus was task-relevant, but this increased automaticity did not lead to detection of conflict when the auditory stimulus was fully task irrelevant.

## Experiment 2: does detection of conflict depend on task relevance of the stimulus or its individual features?

The experimental design of the first experiment rendered the auditory features to be located at the extreme ends of the scale of task relevance, that is, either the conflicting features were task-relevant and the conflicting features were consistently mapped to specific responses, or the conflicting features were task-irrelevant and the conflicting features were not mapped to responses. However, to further understand the relationship between the relevance of the conflicting features and the overlap with responses, we performed a second experiment containing four behavioral tasks. For this second experiment, we recruited 24 new participants. We included two auditory conflicting tasks, similar to content discrimination task I. In one of the auditory tasks (from hereon: content discrimination task II, *Figure 3A*), participants again had to respond according to the content of the auditory stimulus, whereas in the other auditory task (from hereon: location discrimination task, *Figure 3B*) they were instructed to report from which side the auditory stimulus was presented (i.e., left or right speaker).

Furthermore, we included two new tasks in which the conflicting features (location and content) were not task-relevant and participants responded to a non-conflicting feature that was part of the conflicting stimulus (from hereon: volume oddball detection task, *Figure 3C*) or the auditory stimulus was task-irrelevant but its features - location and content- overlapped with the responses to be given (from hereon: horizontal RDM task, *Figure 3D*). The horizontal RDM task was similar to vertical RDM task of experiment 1; however, the dots were now moving on a horizontal plane. In other words, participants were instructed to classify the overall movement of moving dots to either the left or the right. As this is a visual paradigm, the simultaneously presented auditory stimuli are fully task-irrelevant. However, both features of conflict, the content (i.e., 'left' and 'right') and the location (i.e., left and right speaker), of the auditory stimuli could potentially interfere with participants' responses on the visual task, thereby inducing a crossmodal Stroop-like type of conflict (*Stroop, 1935*).

In the volume oddball detection task, participants were presented with the same auditory stimuli as before; however, one out of eight stimuli (12.5%) was presented at a lower volume. Participants were instructed to detect these volume oddballs by pressing the spacebar with their right hand as fast as possible. If they did not hear an oddball, they were instructed to withhold from responding. In this task, theoretically, the selection of an object's feature (e.g., volume) could lead to the selection of all of its features (e.g., sound content, location), as suggested by theories of object-based attention (*Chen, 2012*). This in turn may lead to conflict detection, even if the conflicting features are task-irrelevant. Similar to experiment 1, we included behavioral training in conflict-inducing tasks to inspect if enhanced automaticity of conflict processing would affect conflict detection under task-irrelevant sensory input. Participants performed 500 trials of the volume oddball detection task twice at the very beginning of a session and at the end of a session (*Figure 3E*). During the first run of the task, neither sound content nor sound location was related to any behavioral responses, whereas during the second run these features might have acquired some intrinsic relevance through training on the other tasks. Furthermore, repeated exposure to conflict may prime the conflict monitoring system to exert more control over sensory inputs necessary for more efficient conflict detection, even when these sensory inputs are not task-relevant within the context of the task the participant is performing at that time.

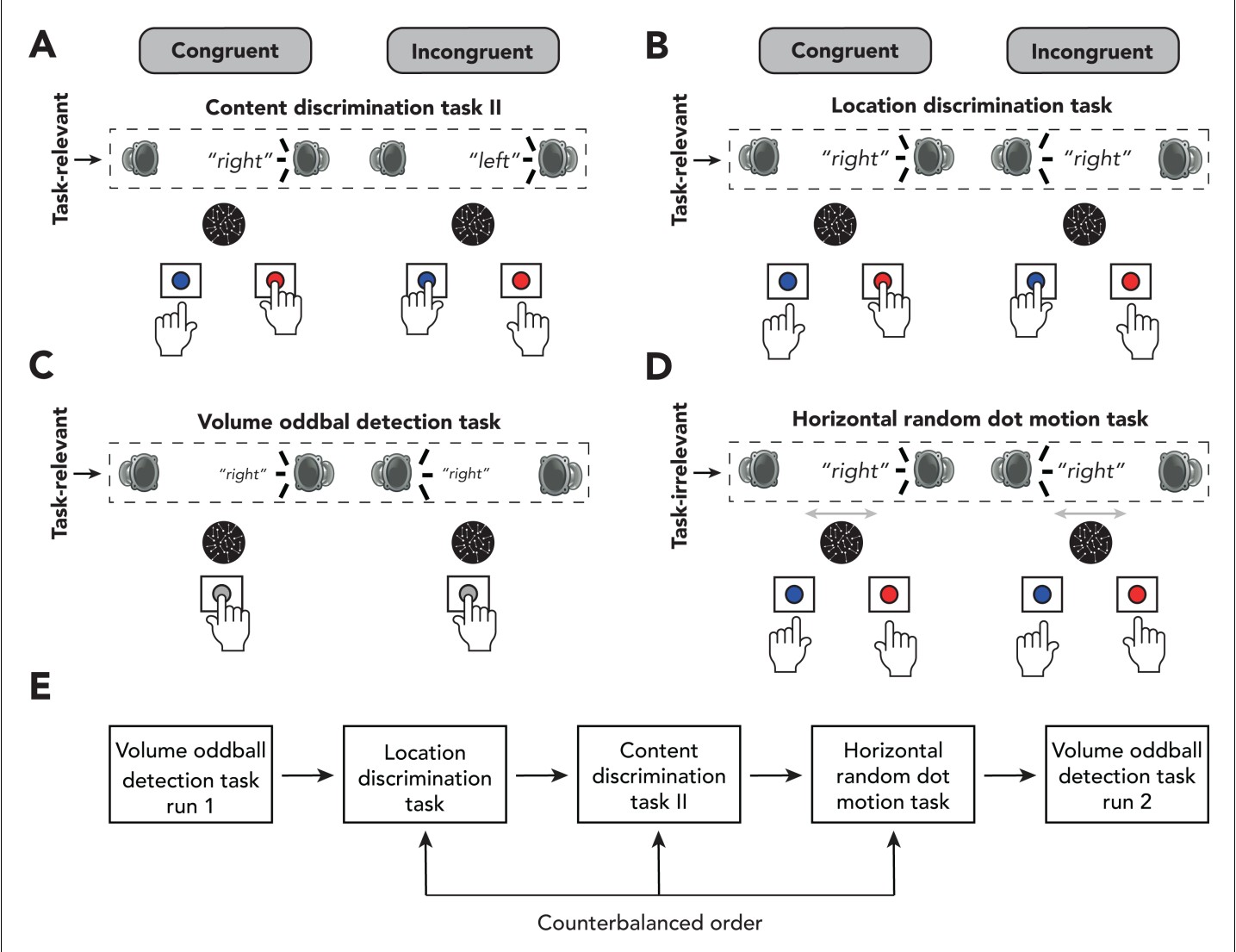

**Figure 3.** Experimental design of experiment 2. (A–D) Schematic representation of the experimental design for auditory content discrimination task II (A), location discrimination task (B), volume oddball task (C), and horizontal random dot-motion (RDM) task (D). In all tasks, the spoken words 'left' and 'right' were presented through either a speaker located on the left or right side of the participant. (A) In auditory content discrimination task II, participants were instructed to report the content ('left' or 'right') of an auditory stimulus via a button press with their left or right hand, respectively, and to ignore the location of the auditory stimulus that was presented. (B) In the auditory location discrimination task, participants were instructed to report the location (left or right speaker) of an auditory stimulus via a button press with their left or right hand, respectively, and to ignore the content of the auditory stimulus that was presented. (C) During the volume oddball task, participants were instructed to detect auditory stimuli that were presented at a lower volume than the majority of the stimuli (i.e., oddballs) by pressing the spacebar with their right hand. (D) In the horizontal RDM, participants were instructed to report the overall movement of dots (left or right) via a button press with their left and right hands, respectively, whilst still being presented with the auditory stimuli. In all four tasks, content of the auditory stimuli could be congruent or incongruent with its location of presentation (50% congruent/incongruent trials). (E) Order of behavioral tasks in experiment 2. Participants always started with the volume oddball task, followed by the location discrimination task, content discrimination task, and horizontal RDM, in randomized order. Participants ended with another run of the volume oddball task.

In order to keep sensory input similar, moving dots (coherence: 0) were presented on the monitor during content discrimination task II, the location discrimination task, and the volume oddball detection task, but these could be ignored. Again, EEG was recorded while participants performed these tasks in order to see if auditory conflict was detected when the auditory stimulus or its conflicting features (i.e., location and content) were task-irrelevant. We performed the same artifact rejection procedure as in experiment 1. For one participant, on average 64.5% (SD = 9.9%) of all epochs

within each task were removed in this procedure, which is 3.9 standard deviations from the average ratio of removed epochs in this experiment (M = 10.3%, SD = 13.9%). Therefore, this participant was excluded from the EEG analysis of experiment 2, resulting in N = 23 for the analysis of EEG data.

## Experiment 2: behavioral effects of conflict only for task-relevant auditory sensory input

Mean RT in the location discrimination task was 338.2 ms (SD = 112.7 ms) and mean ER was 4.5% (SD = 3.2%). For content discrimination task II, RTs were on average 364.0 ms (SD = 127.4 ms) and ERs were 5.5% (SD = 5.8%). For the horizontal RDM, RTs were on average 362.5 ms (SD = 116.5 ms) and ERs were 27.7% (SD = 5.2%). Mean RTs of the volume oddball task were calculated for correct trials in which a response was made (i.e., hit trials) and was 504.5 ms (SD = 178.7 ms, on hit trials). On average, participants had 40.5 hits (SD = 12.8) out of 61.8 oddball trials (SD = 8.6), per run of 500 trials. We will first discuss the behavioral results of the content discrimination, location discrimination, and horizontal RDM tasks as behavioral performance for the volume oddball task is represented in perceptual sensitivity (d'), rather than ER.

rm-ANOVAs (3 × 2 factorial) were performed on mean RTs and ERs from these three tasks, with the factors (1) task and (2) congruency of the auditory features. Again, congruency always relates to the combination of the auditory stimulus features sound content ('left' vs. 'right') and sound location (left speaker vs. right speaker). We observed that participants were slower and made more errors on incongruent trials as compared to congruent trials (RT: $F(1,23) = 75.41$, p<0.001, $\eta_p^2 = 0.77$; ER: $F(1,23) = 68.00$, p <0.001, $\eta_p^2 = 0.75$; *Figure 4A*). This conflict effect was modulated by task (RT: $F(1.55,35.71) = 22.80$, p<0.001, $\eta_p^2 = 0.50$; ER: $F(1.58, 36.36) = 10.18$, p<0.001, $\eta_p^2 = 0.31$) and post-hoc paired samples t-tests (incongruent – congruent) showed that conflict effects were only present in tasks where one of the conflicting features was task-relevant (location discrimination task: $RT_{loc}$: $t(23) = 5.03$, p<0.001, $d = 1.03$; $ER_{loc}$: $t(23) = 6.25$, p<0.001, $d = 1.28$; content discrimination task II: $RT_{cont(II)}$: $t(23) = 8.95$, p<0.001, $d = 1.83$; $ER_{cont(II)}$: $t(23) = 5.93$, p<0.001, $d = 1.21$; horizontal RDM task: $RT_{HRDM}$: $t(23) = 1.44$, p=0.16, $d = 0.29$, $BF_{01} = 1.88$; $ER_{HRDM}$: $t(23) = 1.65$, p=0.11, $d = 0.34$, $BF_{01} = 1.44$). Although in the horizontal RDM task conflict between sound content and location did not affect the speed of responses, stimulus content and location *in isolation* could have potentially interfered with behavioral performance given the overlap of these features with both the plane of dot direction (left/right) and the response scheme (left/right hand). Indeed, trials containing conflict between sound location and dot direction resulted in slower RTs and increased ERs (RT: $t(23) = 2.12$, p=0.045, $d = 0.44$; ER: $t(23) = 5.94$, p<0.001, $d = 1.21$). Similar effects were observed for trials where sound content conflicted with the dot direction, but onlyin ERs (RT: $t(23) = 1.72$, p=0.10, $d = 0.35$, $BF_{01} = 2.85$; ER: $t(23) = 5.55$, p<0.001, $d = 1.13$). This shows that sound content and location *in isolation*, even though both features were task-irrelevant, interfered with task performance.

For the volume oddball task, we tested the effect of auditory congruency on RTs of trials which, by virtue of task instruction, only covers oddball trials in which a correct response was made (i.e., hits). rm-ANOVAs (2 × 2 factorial) with the factors run number and feature congruency revealed no effects of auditory conflict on RT ($F(1,23) = 2.78$, p=0.11, $\eta_p^2 = 0.10$, $BF_{01} = 3.34$; *Figure 4A*, no effects of training, see *Figure 4—figure supplement 1A*). To test whether *individual* features of the auditory stimulus interfered with right-hand responses, we performed an additional 2 × 2 factorial rm-ANOVA with sound content and location as factors. Auditory content significantly affected RTs, whereas sound location did not (content: $F(1,23) = 25.41$, p<0.001, $\eta_p^2 = 0.53$; location: $F(1,23) = 0.99$, p=0.33, $\eta_p^2 = 0.04$, $BF_{01} = 3.90$). Specifically, RTs were slower for the spoken word 'left' (incongruent with the responding hand, M = 528.6 ms, SD = 193.1 ms) as compared to the spoken word 'right' (congruent with the responding hand, M = 493.3 ms, SD = 174.9 ms), revealing interference of sound content *in isolation* on right-hand responses, similar to the horizontal RDM task. Although conflict between the auditory features was not present in RTs, we did observe that sensitivity (d') increased for incongruent (M = 2.66, SD = 0.77) compared to congruent trials (M = 2.11, SD = 0.81, $F(1,23) = 45.62$, p<0.001, $\eta_p^2 = 0.67$; *Figure 4A*). These results show that volume oddball detection performance increases on trials that contain conflict between sound content and location. This effect of conflict on behavioral performance can already be found in the first run, when sound content and location had not yet been related to any responses/task and were thus fully task-irrelevant ($t(23) = $

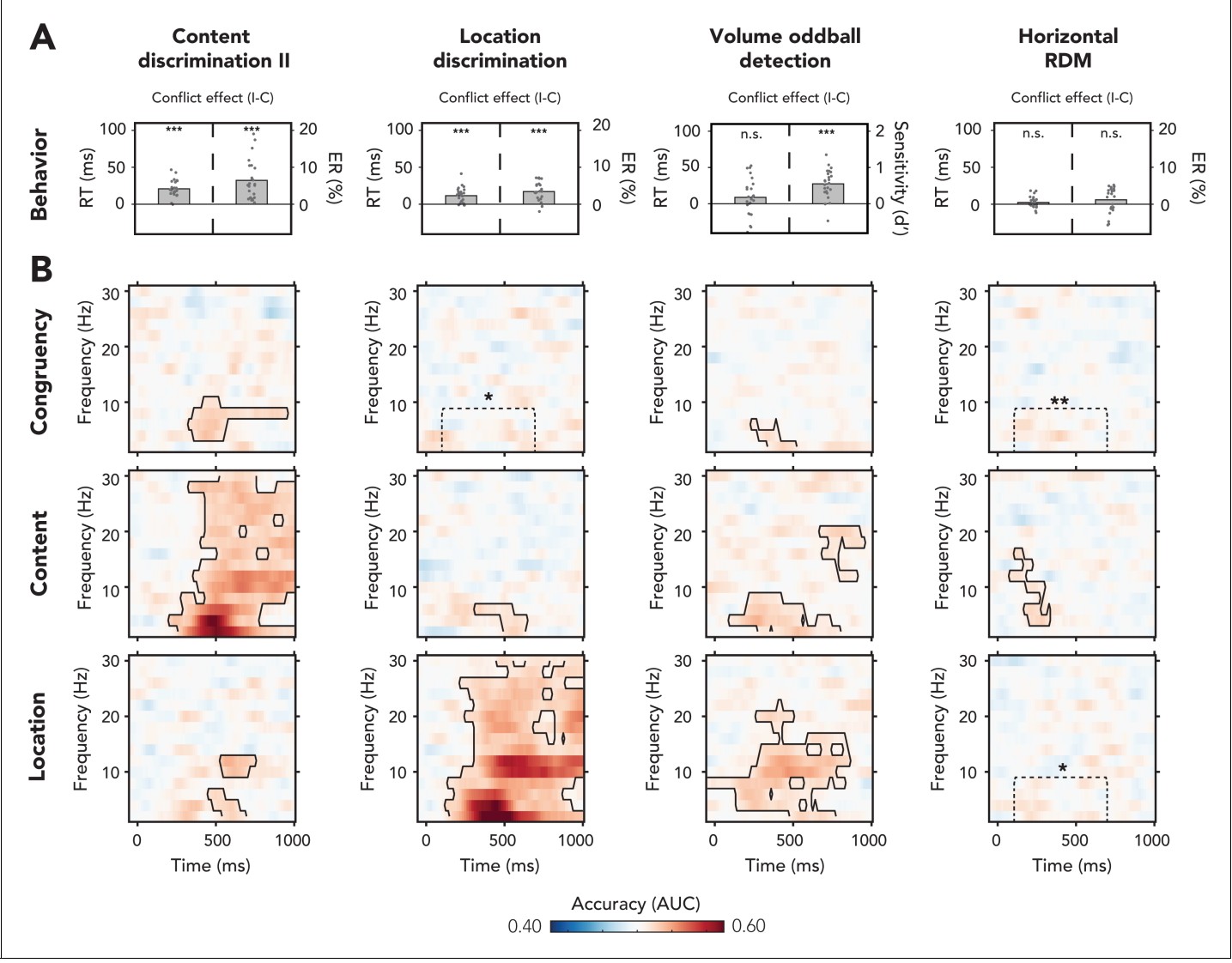

**Figure 4.** Behavioral and multivariate decoding results for experiment 2. (**A, B**) The four columns show data belonging to, from left to right, content discrimination task II, the location discrimination task, the volume oddball detection task, and the horizontal random dot-motion (RDM) task. (**A**) Behavioral results are plotted as conflict effects (incongruent – congruent). Effects of conflict were present in all tasks where the auditory stimulus was task-relevant (content discrimination task II, location discrimination task, and volume oddball). In both auditory discrimination tasks, we observed longer reaction times (RTs) (left bar) and increased error rates (right bar) for incongruent compared to congruent trials. For the volume oddball, we did not observe an effect in RT, but increased sensitivity (d') on incongruent compared to congruent trials. Dots represent individual participants. The data that is shown here can be found in *Figure 4—source data 1*. (**B**) Multivariate classifier accuracies for different stimulus features (auditory congruency, auditory content, and auditory location). Classifier accuracies (area under the curve [AUC]) are plotted across a time-frequency window of −100 ms to 1000 ms and 2–30 Hz. Classifier accuracies are thresholded (cluster-based corrected, one-sided: $\bar{X}>0.5$, p<0.05), and significant clusters are outlined with a solid black line. The dotted box shows the predefined ROI on which we performed a hypothesis-driven analysis. Note that the data shown for the volume oddball task was merged over both runs. *p<0.05, **p<0.01, ***p<0.001; n.s.: p>0.05.

The online version of this article includes the following source data and figure supplement(s) for figure 4:

**Source data 1.** Behavioral results of experiment 2.

**Figure supplement 1.** Effects of exposure to conflict inducing task on behavioral effects of conflict and decoding performance in the volume oddball task of experiment 2.

**Figure supplement 1—source data 1.** Behavioral results of the volume oddball task - first and second run.

**Figure supplement 2.** Sensory feature decoding in the time-domain.

5.71, p<0.001, $d$ = 1.17). There was no significant interaction between run number and auditory stimulus congruency ($F(1,23)$ = 1.40, p=0.25, $\eta_p^2$ = 0.06, $BF_{01}$ = 2.13; *Figure 4—figure supplement 1B*; hit rates and false alarms are plotted in this figure supplement as well).

## Experiment 2: detection of conflict occurs when any feature of a stimulus is related to the response on the primary task or is task-relevant (even a non-conflicting feature)

We again trained multivariate classifiers on single-trial time-frequency data to test whether the auditory stimulus features (i.e., content, location, and congruency) were processed when (1) the auditory conflicting features were task-relevant and overlapped with the response-mapping (content and location discrimination tasks), (2) the auditory conflicting features were task-irrelevant and another feature of the conflicting stimulus was task-relevant (volume oddball task), or when (3) the auditory conflicting features were task-irrelevant, but its conflicting features overlapped with the response-mapping in the task (horizontal RDM task).

Cluster-based analyses across the entire T-F range revealed neural signatures of conflict processing in the theta-band when the content of the auditory stimulus was task-relevant (content discrimination task II: p<0.001, one-sided: $\bar{X}$>0.5, cluster-corrected; frequency range: 4–10 Hz, peak frequency: 4 Hz, time range: 328–953 ms, peak time: 438 ms; *Figure 4B*) and when the volume of the auditory stimulus was task-relevant (volume oddball task: p=0.03, one-sided: $\bar{X}$>0.5, cluster-corrected; frequency range: 2–6 Hz, peak frequency: 2 Hz, time range: 234–516 ms, peak time: 438 ms; *Figure 4B*, see *Figure 4—figure supplement 1E*: no training effects for the volume oddball task). Both observations of congruency decoding are in line with the presence of conflict-related behavioral effects in these tasks (*Figure 4A*). No significant clusters of above-chance classifier accuracy were found after correcting for multiple comparisons in the location discrimination task and the horizontal RDM task (*Figure 4B*). However, a hypothesis-driven analysis focused on the post-stimulus theta-band (2–8 Hz, 100–700 ms) revealed that congruency decoding accuracies within this ROI were significantly above chance for both tasks as well (location discrimination: $t(22)$ = 2.00, p=0.03, $d$ = 0.42; horizontal RDM: $t(22)$ = 2.89, p=0.004, d = 0.60). Thus, we observed that conflict of the auditory stimulus is detected when one of the auditory conflicting features is task-relevant (content and location discrimination tasks), when one of its non-conflicting features is task-relevant (volume oddball task), and when none of the auditory features is task-relevant but these features overlap with the response-mapping of the task (horizontal RDM task). To qualify the differences between tasks, we combine the data from all experiments and compare effect sizes across tasks at the end of this section (*Figure 5*).

## Experiment 2: sensory features are processed in all tasks

Next, we trained classifiers to distinguish trials based on sound location and content in order to inspect sensory processing. We found neural signatures of the processing of sound content for all four tasks: content discrimination II (p<0.001, one-sided: $\bar{X}$>0.5, cluster-corrected; frequency range: 2–30 Hz, peak frequency: 4 Hz, time range: 203–1000 ms, peak time: 469; *Figure 4B*), location discrimination (p=0.01, one-sided: $\bar{X}$>0.5, cluster-corrected; frequency range: 2–6 Hz, peak frequency: 2 Hz, time range: 313–641 ms, peak time: 563 ms; *Figure 4B*), horizontal RDM task (p<0.001, one-sided: $\bar{X}$>0.5, cluster-corrected; frequency range: 4–16 Hz, peak frequency: 4 Hz, time range: 78–328 ms, peak time: 281 ms; *Figure 4B*). For the volume oddball task, we observed a delta/theta-band cluster and a late beta-band cluster (delta/theta: p<0.001, one-sided: $\bar{X}$>0.5, cluster-corrected; frequency range: 2–8 Hz, peak frequency: 4 Hz, time range: 94–797 ms, peak time: 281 ms; late beta: p=0.01, one-sided: $\bar{X}$>0.5, cluster-corrected; frequency range: 12–20 Hz, peak frequency: 20 Hz, time range: 672–953 ms, peak time: 828 ms; *Figure 4B*).

Furthermore, sound location could be decoded from the content discrimination task II (delta/theta: p=0.02, one-sided: $\bar{X}$>0.5, cluster-corrected; frequency range: 2–6 Hz, peak frequency: 2 Hz, time range: 453–688 ms, peak time: 609 ms; alpha: p=0.03, one-sided: $\bar{X}$>0.5, cluster-corrected; frequency range: 10–12 Hz, peak frequency: 12 Hz, time range: 531–750 ms, peak time: 578 ms; *Figure 4B*), the location discrimination task (p<0.001, one-sided: $\bar{X}$>0.5, cluster-corrected; frequency range: 2–30 Hz, peak frequency: 2 Hz, time range: 109–1000 ms, peak time: 453 ms; *Figure 4B*), and the volume oddball task (p<0.001, one-sided: $\bar{X}$>0.5, cluster-corrected;

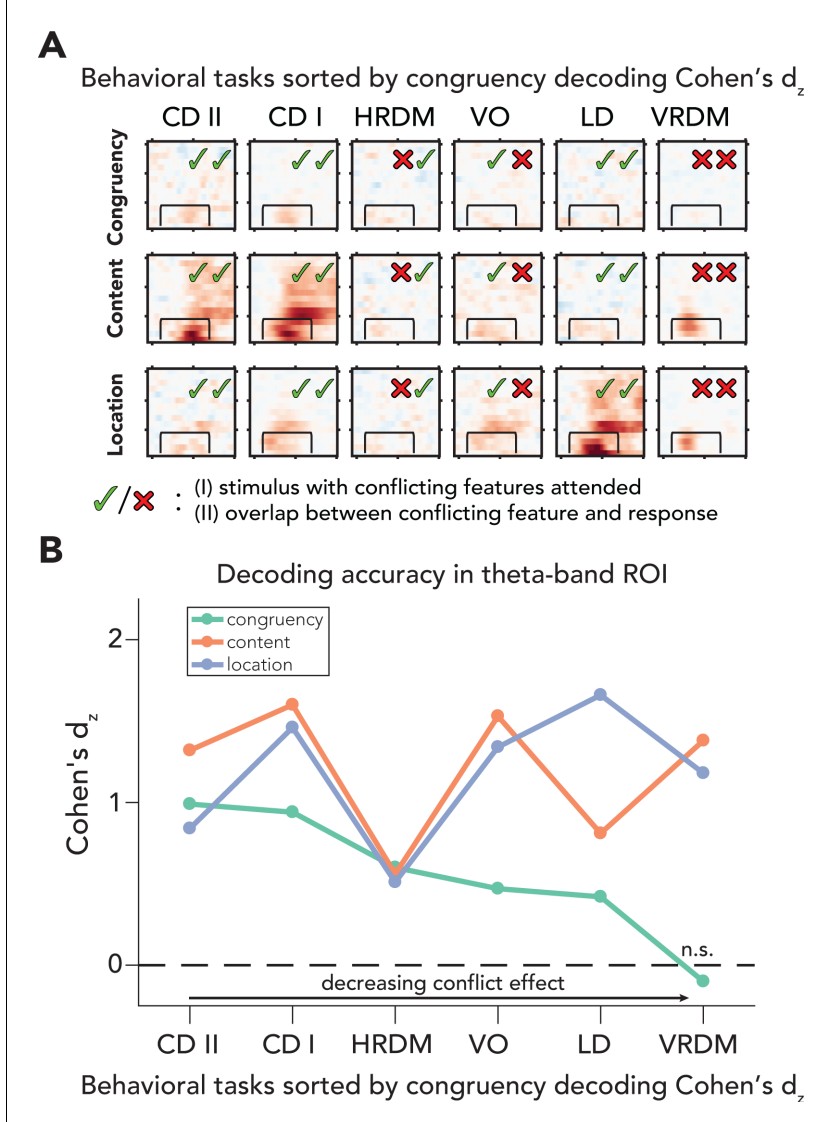

**Figure 5.** Processing of sensory and conflict features for different levels of task relevance. (**A**) Summary of the decoding results of all behavioral tasks, sorted by congruency decoding effect size (Cohen's $d_z$) in a preselected time-frequency ROI. The data in these plots are identical to the ones shown in *Figures 2* and *4*. (**B**) Effect sizes are shown for all task/feature combinations derived from a predefined ROI (2–8 Hz and 100–700ms) and sorted according to effect size of congruency decoding. Effect sizes for congruency decoding were dependent on behavioral task (downward slope of the green line), whereas this was not the case, or less so, for the decoding of content and location. The data can be found in *Figure 5—source data 1*. CD II: content discrimination task II; CD I: content discrimination task I; HRDM: horizontal RDM task; VO: volume oddball detection task; LD: location discrimination task; VRDM: vertical RDM task; n.s.: p>0.05.

The online version of this article includes the following source data and figure supplement(s) for figure 5:

**Source data 1.** Decoding results within ROI for all tasks.

**Figure supplement 1.** Topographic maps of reconstructed activation patterns and effects sizes for alternative ROIs.

frequency range: 2–22 Hz, peak frequency: 10 Hz, time range: −47 ms to 891 ms, peak time: 469 ms; *Figure 4B*). Initially, we did not observe a significant cluster of location decoding in the horizontal RDM, although the hypothesis-driven analysis revealed that accuracies within the predefined theta-band ROI were significantly above chance level as well ($t(23) = 2.47$, p=0.01, $d = 0.51$).

One aspect of these results is worth highlighting. When participants responded to the location of the auditory stimulus, location decoding revealed a broadband power spectrum, similar to sound content decoding when sound content was task-relevant (content discrimination tasks). This broad frequency decoding may be due to the fact that these features were task-relevant, but these results may also partially reflect response preparation and response execution processes as these auditory features were directly associated with a specific response. In order to test whether the earliest sensory responses were already modulated by task relevance and to link this to previous event-related potential (ERP) studies (*Alilović et al., 2019*; *Molloy et al., 2015*; *Woldorff et al., 1993*), we performed an additional time-domain multivariate analysis on these sensory features (T-F analyses are not well suited to address questions about the timing of processes). Because we were interested in the earliest sensory responses, we performed this analysis on data from experiment 2 only as all task parameters were best matched (e.g., in all tasks, a visual stimulus was presented, no training, etc.). We observed increased decoding for task-relevant sensory features compared to task-irrelevant features, starting ~250 ms (sound location RT: $M$ = 338 ms) and ~330 ms (sound content RT: $M$ = 364 ms) after stimulus presentation (*Figure 4—figure supplement 2*). The onset of these differences starts before a response is made, which may suggest that sensory processing of these features is indeed affected by task relevance; however, processes building up towards motor execution, such as decision-making and response preparation processes, cannot be excluded as potential factors driving the higher decoding accuracies in tasks where specific features are task-relevant and hence correlated with decision and motor processes. These results are elaborated upon in the Discussion.

In conclusion, in line with the behavioral results, we observed that the processing of conflict between two stimulus features (i.e., location and content of an auditory stimulus) was present in all tasks of experiment 2. This indicates that conflict can be detected when one of the auditory conflicting features is task-relevant (content and location discrimination tasks), when one of its non-conflicting features is task-relevant (volume oddball task), and when there is overlap in the response-mapping with any of its task-irrelevant conflicting features (horizontal RDM task). Overall, this reveals that when the conflicting stimulus itself is attended or when its conflicting features overlap with the response scheme, all of its features seem to be processed and integrated to elicit conflict.

## All experiments: decreasing task relevance hampers cognitive control operations, but not sensory processing

The neural data from all six different tasks over two experiments suggests that if sensory input is task-irrelevant, processing of that information is preserved, while cognitive control operations are strongly hampered (*Figures 2B, 4B,* and *5B*). To quantify this observation, we calculated Cohen's $d_z$ for all tasks and features (based on the preselected ROI), sorted tasks according to the effect sizes of congruency decoding, and plotted Cohen's $d_z$ across tasks for all features (conflict, content, and location; *Figure 5*). For each feature and task, we extracted individual classifier area under the curve (AUC) values and performed an analysis of covariance (ANCOVA) on these accuracies, with fixed effects being task and stimulus feature. We found main effects for behavioral task and stimulus feature (task: $F(5,402)$ = 17.25, p<0.001, $\eta_p^2$ = 0.18; stimulus feature: $F(2,402)$ = 44.61, p<0.001, $\eta_p^2$ = 0.18). Crucially, the interaction between task and stimulus feature was also significant ($F(10,402)$ = 18.80, p<0.001, $\eta_p^2$ = 0.32), showing that the accuracy of conflict decoding decreased more across tasks as compared to content and location decoding. We next performed t-tests (one-sided, $\bar{X}$ > 0.5) on ROI accuracies from every task/feature combination to assess decoding performance for all stimulus features for the different task-relevance manipulations (see *Figure 5—source data 1*). We observed that congruency decoding accuracies were strongly influenced by task, whereas this was not the case for decoding accuracies of stimulus content and location (*Figure 5B*). Note that these results were robust and not dependent on the specific ROI that was selected because using other ROI windows led to similar patterns of results, that is, decreased congruency decoding under task-irrelevant input, but relatively stable sensory feature decoding (*Figure 5—figure supplement 1C, D*). Classifier weights were extracted from the ROI for all tasks and features, transformed to activation patterns and plotted in topomaps, to show the patterns of activity underlying the decoding results (*Figure 5—figure supplement 1A*).

## Discussion

Although it has been hypothesized for a long time that only basic physical properties of task-irrelevant sensory input are processed (*Treisman and Gelade, 1980*), over the past few years an overabundance of processes has been found to be preserved in the absence of attention (*Fahrenfort et al., 2017*; *Li et al., 2002*; *Peelen et al., 2009*). Here, we aimed to push the limits of the brain's capacity to process unattended information and addressed whether cognitive control networks can be recruited when conflicting features of sensory input are task-irrelevant. Interestingly, similar cognitive control functions have been shown to occur when stimuli are masked and hence conscious awareness is strongly reduced (*Atas et al., 2016*; *D'Ostilio and Garraux, 2012b*; *Huber-Huber and Ansorge, 2018*; *Jiang et al., 2015b*; *Jiang et al., 2018*; *van Gaal et al., 2008*; *van Gaal et al., 2011*).

In two omnibus experiments with six different tasks, we presented stimuli with potentially auditory-spatial conflicting stimulus features (e.g., the word 'left' presented on the right side) to participants, whilst they were performing several behavioral tasks. These tasks manipulated whether the features sound content and location of the auditory stimulus were task-relevant and whether these features were mapped to specific overlapping responses of the primary task. We observed clear signals of conflict processing in behavior (i.e., longer RTs, increased ERs, increased sensitivity) and brain activity (i.e., above-chance decoding accuracy in the theta-band) when the conflicting features of the auditory stimulus were task-relevant, that is, in the content and location discrimination tasks, when another non-conflicting feature of the auditory stimulus was task-relevant, but the conflicting features content and location were not (volume oddball task) and when the conflicting features were not task-relevant but when they overlapped with the response scheme of the task (horizontal RDM task). When the features of the auditory stimulus were task-irrelevant and orthogonal to the response scheme of the primary task, that is, in the vertical RDM task, we did not observe any effects of conflict in behavior or neural measures. The absence of conflict effects was supported by Bayesian analyses, showing reliable evidence in favor of the null hypothesis. Strikingly, the individual stimulus features, that is, stimulus location/content, were always processed, regardless of their task relevance and response relevance. Note that this dissociation – hampered conflict processing yet preserved perceptual processing – cannot be explained by a lack of statistical power because decoding accuracy of stimulus location/content was comparable between behavioral tasks. These results highlight that relatively basic stimulus properties escape the attentional bottleneck, lending support to previous studies (e.g., *Fahrenfort et al., 2017*; *Li et al., 2002*; *Peelen et al., 2009*; *Sand and Wiens, 2011*; *Treisman and Gelade, 1980*), but furthermore showcase that an attentional bottleneck for detecting conflict (integration of stimulus features) exists upstream in the hierarchy of cognitive processing. Below we link the observed results to the existing literature.

### Object-based attention as a prerequisite for feature integration (leading to conflict)

So why are auditory content and location not integrated to form conflict when all of the auditory features are task-irrelevant? It has been suggested that the MFC is crucial for monitoring the presence of conflict, through the detection of coinciding inputs (*Cohen, 2014*). In our paradigm, it thus seems crucial that information related to the auditory features content and location reaches the MFC to be able to detect conflict, although control networks can undergo reconfiguration under certain circumstances (*Canales-Johnson et al., 2020*). Previous studies have shown that task-irrelevant stimuli can still undergo elaborate processing (*Li et al., 2002*; *Peelen et al., 2009*). Our decoding results show that task-irrelevant features are indeed processed by the brain (*Figures 2B*, *4B,* and *5*). Interestingly, conflict between two task-irrelevant features was detected when another feature of the conflicting stimulus was task-relevant (volume oddball task) or when the conflicting features had overlap with the overall response scheme (horizontal RDM task), but remained undetected when none of the auditory features was task-relevant and there was no overlap with the response scheme (vertical RDM task). We argue that this difference is due to the fact that in the volume oddball and horizontal RDM tasks, the task-irrelevant conflicting features were selected through object-based attention. Theories of object-based attention have suggested that when one stimulus feature of an object is task-relevant and selected, attention 'spreads' to all other features of the attended stimulus, even when these features are task-irrelevant or part of a different stimulus or modality (*Chen, 2012*;

*Chen and Cave, 2006*; *O'Craven et al., 1999*; *Turatto et al., 2005*; *Wegener et al., 2014*; *Xu, 2010*). In the volume oddball task, a non-conflicting feature of the auditory stimulus (volume) was task-relevant, but this allowed for the selection of the other task-irrelevant features through object-based attention. In the horizontal RDM task, on the other hand, the conflicting features of the task-irrelevant auditory stimulus overlapped with the overall response scheme or task-set of the participant, namely discriminating rightward- versus leftward-moving dots. This may have led to the automatic classification of all sensory input according to this task-set (as either coding for 'left' or 'right'), even when that input was not relevant for the task at hand. Possibly, through this classification, attentional resources could be exploited for the processing of these task-irrelevant features. This is especially interesting because in all conflict analyses incongruency was defined as the mismatch between the two features of the auditory stimulus (location and sound) and not between a visual feature (leftward-moving dots) and one feature of the auditory stimulus (e.g., the word 'right'). Note that we report additional behavioral results that show clear indications of conflict when the task-relevant feature of the visual stimulus interferes directly with a *single* task-irrelevant feature of the auditory task (e.g., auditory content-dot-motion conflict).

When inspecting the T-F maps for the vertical RDM task, the relatively fleeting temporal characteristics of the processing of the task-irrelevant stimulus features (sound content and location) might suggest that the integration of these features may not be possible due to a lack of time as proposed in the incremental grouping model of attention (*Roelfsema, 2006*; *Roelfsema and Houtkamp, 2011*). However, the time window in which conflict was decodable when the auditory conflicting features were task-relevant coincides with the time range in which these features could be decoded when the auditory conflicting features were task-irrelevant (*Figure 5A*). Therefore, it seems unlikely that the more temporally constrained processing of task-irrelevant stimulus features is the cause of hampered conflict detection. Besides time being a factor, the processing of task-irrelevant features in the vertical RDM task may have also been too constrained to (early) sensory cortices and therefore could not progress to integration networks, including the MFC, necessary for the detection of conflict. Speculatively, the processing of task-irrelevant auditory features was relatively superficial due to the relatively few remaining resources (*Lavie et al., 2004*; *Sigman and Dehaene, 2006*; *Zylberberg et al., 2010*; *Zylberberg et al., 2011*), and combined with a lack of object-based attention, this may have prevented the propagation of the information to the MFC. It has been hypothesized that unattended (sometimes referred to as 'preconscious'; *Dehaene et al., 2006*; *Dehaene and Changeux, 2011*) stimuli are not propagated deeply in the brain, but still allow for shallow recurrent interactions in sensory cortices. The poor spatial resolution of EEG measurements and the specifics of our experimental setup, however, do not allow to test these ideas regarding the involvement of spatially distinct cortices. Yet, previous work of our group suggests that task-irrelevant nonconscious information does not propagate to the frontal cortices, whereas task-relevant nonconscious information does. We demonstrated that masked task-irrelevant conflicting cues induced similar early processing in sensory cortices as compared to masked task-relevant cues, but prohibit activation of frontal cortices (*van Gaal et al., 2008*). These findings are not conclusive, and so we believe that uncovering the role of task relevance in processing of (nonconscious) information deserves more attention in future work (see also *van Gaal et al., 2012* for a discussion on this issue).

## Sensory processing is weakened, but conflict processing hampered in the absence of task relevance

We show that conflict processing is absent when conflicting features are fully task-irrelevant, while evidence of sensory processing is still present in neural data (*Figures 2B*, *4B,* and *5*). Even though sensory processing of auditory features seems relatively preserved under various levels of task relevance of these features, it appears that sensory processing may in fact also be affected when the feature is task-irrelevant (*Figure 4—figure supplement 2*), in line with previous studies (e.g., *Alilović et al., 2019*; *Jehee et al., 2011*; *Kok et al., 2012*; *Kouider et al., 2016*), although to a lesser extent than conflict processing (*Figure 5B*). For example, when sound location is the task-relevant feature (i.e., in the location discrimination task), decoding accuracies for that feature are more broadband in the frequency domain (*Figure 4B*) and higher in the time domain (*Figure 4—figure supplement 2*), compared to location decoding performance in other tasks. This increased decoding accuracy is present even before a response has been made, suggesting decreased early stage sensory processing in tasks where the decoded feature is not task-relevant. However, despite sensory

processing being weakened under decreasing levels of task relevance, it is not diminished, in line with previous findings of ongoing processing in the (near) absence of attention (*Fahrenfort et al., 2017*; *Li et al., 2002*; *Peelen et al., 2009*). Processing of conflict between the two interfering auditory features, on the contrary, is hampered when the features are fully task-irrelevant. This is further supported by the significant interaction between task and feature in terms of decoding performance within the predefined ROI (*Figure 5B*). Summarizing, although processing of sensory features is degraded under decreasing levels of task relevance it is present regardless of attention, whereas detection of conflict between these features is no longer possible when the features are fully task-irrelevant.

Besides object-based attention, the process through which attentional resources are allocated to the processing of task-irrelevant features of a task-relevant object, other mechanisms might also play a role in the extent to which sensory information is processed, such as the active suppression of task-irrelevant information. It has been shown that task-irrelevant information that is response-relevant, and can thus potentially interfere with performance on the primary task, can be suppressed to minimize interference (*Appelbaum et al., 2011*; *Janssens et al., 2018*; *Polk et al., 2008*; but see *Egner and Hirsch, 2005*). This would result in more reduced sensory processing, indexed by lower decoding performance, for task-irrelevant features that are response-relevant than for task-irrelevant features that are not. Disentangling the effects of such mechanisms, object-based attention and their possible interactions on the processing of sensory and cognitive information, however, falls outside the scope of this work.

## Disentangling effects of conflict and task difficulty

For our main analysis, we trained a multivariate classifier on congruent versus incongruent trials and observed effects of task relevance of the performance of the classifier, that is, decoding performance was hampered when conflicting features were fully task-irrelevant (*Figures 2B*, *4B,* and *5B*). Moreover, we report behavioral effects of conflict in all *auditory* tasks as well (*Figures 2A* and *4A*). Given that behavioral performance on the auditory tasks is worse for incongruent trials as compared to congruent trials, one may wonder whether our multivariate decoder is in fact picking up information related to conflict detection or processes related to task difficulty. Whether medial frontal theta-band oscillations are a reflection of conflict detection or task difficulty, and whether these factors can be dissociated in principle, has been the topic of debate in the literature (*Grinband et al., 2011a*; *Grinband et al., 2011b*; *McKay et al., 2017*; *Ruggeri et al., 2019*; *Yeung et al., 2011*). On the one hand, it has been shown that activity in the dorsal medial prefrontal cortex is related to RT, suggesting that neural markers of conflict may in fact reflect time on task (*Grinband et al., 2011b*; *Grinband et al., 2011b*; *Ruggeri et al., 2019*). On the contrary, other research has shown that enhanced prefrontal theta-band oscillations are found in conflicting trials even when controlling for RT (*Cohen and van Gaal, 2014*) or task difficulty (*McKay et al., 2017*). The decoding results presented in this work likely reflect conflict processing, and not just task difficulty, for two reasons. First, the spatial distribution and time-frequency dynamics of the congruency decoding results are comparable to those more commonly found in the literature on conflict processing, even in a study where conflicting signals were matched for RT (*Cohen and van Gaal, 2014*). Specifically, using the content discrimination task of experiment 1 as example, we observe effects of conflict centered on the theta-band and ~230–610 ms post-conflict presentation, with a clear medial frontal spatial profile (*Figure 2B*, *Figure 5—figure supplement 1A*). Second, auditory stimulus conflict was decodable from neural data for two tasks in which there were either no effects of conflict – or task difficulty – on behavioral performance (i.e., horizontal RDM task), or even increased behavioral performance on conflicting trials (i.e., volume oddball task). Therefore, we believe that the observed congruency decoding results presented here are mainly driven by the detection of conflicting sensory inputs and are not, or much less so, driven by task difficulty.

## Conflict between features of a task-irrelevant stimulus versus conflict between stimuli

Contrary to the current study, previous studies using a variety of conflict-inducing paradigms and attentional manipulations reported conflict effects in behavior and electrophysiological recordings induced by unattended stimuli or stimulus features (*Mao and Wang, 2008*; *Padrão et al., 2015*;

*Zimmer et al., 2010*). However, our study deviates from those studies in several crucial aspects. First, we explicitly separate task-relevant stimulus features that cause conflict and task-relevant features that do not, parsing the cognitive components that induce cognitive control in this context. Furthermore, in the RDM and volume oddball tasks we tested whether conflict between two task-irrelevant features could be detected by the brain. Specifically, we investigated if conflict between two task-irrelevant features would be detected in the presence or absence of object-based attention (e.g., volume oddball task vs. vertical RDM task), also manipulating whether task-irrelevant conflicting features mapped onto the response or not (horizontal RDM task vs. vertical RDM task). This approach is crucially different from previous studies that exclusively tested whether a task-irrelevant or unattended stimulus (feature) could interfere with processing of a task-relevant feature (*Mao and Wang, 2008*; *Padrão et al., 2015*; *Zimmer et al., 2010*). Under such conditions, at least one source contributing to the generation of conflict (i.e., the task-relevant stimulus) is fully attended, and therefore, one cannot claim that under those circumstances conflict detection occurs outside the scope of attention.

It can be argued that in our horizontal RDM task the task-irrelevant auditory features (location and content) that mapped onto the response of the primary task could interfere with the processing of horizontal dot-motion, that is, the task-relevant feature. This is in fact true, as we found effects of auditory content-dot-motion and auditory location-dot-motion conflict in behavior (both on RTs and ERs). This highlights that a single feature of a task-irrelevant stimulus can interfere with the response to a task-relevant stimulus when there are overlapping feature-response-mappings. This is different from two features of a task-irrelevant stimulus to produce inherent conflict (e.g., between auditory content and location), which is what we specifically investigated by always testing the presence of auditory content-location conflict only. A similar argument might be made for our vertical RDM and volume oddball tasks because in those cases the auditorily presented stimuli could potentially conflict with responses that were exclusively made with the right hand, for example, the spoken word 'left' or the sound from left location may conflict generally more with a right-hand response (independent of the up/down classification or oddball detection) than the spoken word 'right' or the sound from right location. In the vertical RDM task, the auditorily presented stimuli were truly task-irrelevant as both stimulus content and location in isolation did not affect behavior. In the volume oddball task, sound content and location were task-irrelevant features, but these features were part of the attended stimulus and hence selected through object-based attention. In this task, the content of the auditory stimuli (e.g., 'left') did interfere with right-hand responses to the volume oddball task, resulting in longer RTs (compared to 'right'). Moreover, in this task we did find behavioral and neural effects of conflict between two auditory features (*Figure 4*). The absence of conflict effects in the vertical RDM and presence of such effects in the volume oddball task and horizontal RDM indicates that at least one feature of the stimulus containing the conflicting features should be task-relevant or associated with a response in order for conflict to be detected. Summarizing, we show that the brain is not able to detect conflict that emerges between two features of a task-irrelevant stimulus in the absence of object-based attention.

Lastly, in other studies, conflicting stimuli were often task-irrelevant on one trial (e.g., because they were presented at an unattended location) but task-relevant on the next (e.g., because they were presented at the attended location) (e.g., *Padrão et al., 2015*; *Zimmer et al., 2010*). Such trial-by-trial fluctuations of task relevance allow for across-trial modulations to confound any current trial effects (e.g., conflict-adaptation effect) and also induce a 'stand-by attentional mode' where participants never truly disengage to be able to determine if a stimulus is task-relevant. We prevented such confounding effects in the present study, where the (potentially) conflicting features or the auditory stimulus were task-irrelevant on every single trial in the vertical RDM, horizontal RDM, and volume oddball task.

## Differences between response conflict and perceptual conflict cannot account for absence of conflict detection in task-irrelevant sensory input

One difference between the content and location discrimination tasks, on the one hand, and the volume oddball and RDM tasks, on the other, was the task relevance of the (conflicting) auditory features. Another major difference between these groups of tasks was, consequently, the origin of the conflict. When the auditory stimuli were task-relevant, the origin of conflict was found in the

interference of a task-irrelevant feature on behavioral performance, whereas for the other tasks this was not the case. We argued that in the volume oddball and RDM tasks salient auditory stimuli could be *intrinsically* conflicting. Intrinsic conflict is often referred to as perceptual conflict, as opposed to the aforementioned behavioral conflict (*Kornblum, 1994*). Although perceptual conflict effects are usually weaker than response conflict effects, both in behavior and electrophysiology (*Frühholz et al., 2011*; *van Veen et al., 2001*; *Wang et al., 2014*), this difference in the origin of the conflict is unlikely to explain why we did not observe effects of conflict under task-irrelevant sensory input, as opposed to earlier studies.

First, several neurophysiological studies have previously reported electrophysiological modulations by perceptual conflict centered on the MFC (*Jiang et al., 2015a*; *Nigbur et al., 2012*; *van Veen et al., 2001*; *Wang et al., 2014*; *Zhao et al., 2015*). Second, an earlier study using a task similar to ours (but including only task-relevant stimuli) showed effects of perceptual conflict, that is, unrelated to response-mapping, in both behavior and neural measures (*Buzzell et al., 2013*). Third, the prefrontal monitoring system has previously been observed to respond when participants view other people making errors (*Jääskeläinen et al., 2016*; *van Schie et al., 2004*), suggesting that cognitive control can be triggered without the need to make a response. Fourth, in our volume oddball task, where conflict was perceptual in nature as well, we did observe conflict effects, both in behavior and neural data.

## Inattentional deafness or genuine processing of stimulus features?

The lack of conflict effects in the vertical RDM task might suggest a case of inattentional deafness, a phenomenon known to be induced by demanding visual tasks, which manifests itself in weakened early (~100 ms) auditory evoked responses (*Molloy et al., 2015*). Interestingly, human speech seems to escape such load modulations and is still processed when unattended and task-irrelevant (*Olguin et al., 2018*; *Röer et al., 2017*; *Zäske et al., 2016*), potentially because of its inherent relevance, similar to (other) evolutionary relevant stimuli such as faces (*Finkbeiner and Palermo, 2017*; *Lavie et al., 2003*). Indeed, the results of our multivariate analyses demonstrate that spoken words are processed (at least to some extent) when they are task-irrelevant as stimulus content (the words 'left' and 'right', middle row, *Figures 2B* and *4B*, and *Figure 5B*) and stimulus location (whether the word was presented on the left or the right side', bottom row, *Figures 2B* and *4B*, and *Figure 5B*) could be decoded from time-frequency data for all behavioral tasks. For all tasks, classification of stimulus content was present in the theta-band (4–8 Hz), which is in line with a previously proposed theoretical role for theta oscillations in speech processing, namely that they track the acoustic envelope of speech (*Giraud and Poeppel, 2012*). After this initial processing stage, further processing of stimulus content is reflected in more durable broadband activity for the content discrimination tasks, possibly related to higher-order processes (e.g., semantic) and response preparation (middle row and left column, *Figures 2B* and *Figure 4B*). Similarly to processing of stimulus content, processing of stimulus location was most strongly reflected in the delta- to theta-range for all tasks (*Figure 4B*), which may relate to the auditory N1 ERP component, an ERP signal that is modulated by stimulus location (*Fuentemilla et al., 2006*; *Lewald and Getzmann, 2011*; *Salminen et al., 2015*). We also observed above-chance location decoding in the alpha-band for task-relevant auditory stimuli, convergent with the previously reported role of alpha-band oscillations in orienting and allocating audio-spatial attention (*Weisz et al., 2014*).

Thus, the characteristics of the early sensory processing of (task-irrelevant) auditory stimulus features in our study are in line with recent findings of auditory and speech processing. Moreover, our observations are in line with recent empirical findings that suggest a dominant role for late control operations, as opposed to early selection processes, in resolving conflict (*Itthipuripat et al., 2019*). Specifically, this work showed that in a Stroop-like paradigm both target and distractor information is analyzed fully, after which the conflicting input is resolved. Extending on this, we show preserved initial processing of task-irrelevant sensory input, but hampered late control operations necessary to detect conflict, at least in the current setup.

## Automatization of conflict processing does not promote detection of task-irrelevant conflict

Previous studies investigating conflict through masking procedures concluded that conflict detection by the MFC may happen automatically and is still operational under strongly reduced levels of stimulus visibility (*D'Ostilio and Garraux, 2012b*; *Jiang et al., 2015b*; *van Gaal et al., 2008*). Such automaticity can often be enhanced through training of the task. For example, training in a stop-signal paradigm in which stop-signals were rendered unconscious through masking led to an increase in the strength of behavioral modulations of these stimuli (*van Gaal et al., 2009*). In order to see whether enhancing such automaticity could hypothetically increase the likelihood of conflict detection, we included extensive training sessions in the first experiment and had measurements of the volume oddball task before and after exposure to conflicting tasks in the second experiment. In experiment 1, we found no neural effects of conflict detection in the vertical RDM task, even when participants had been trained on the auditory task for 3600 trials (*Figure 2B*, *Figure 2—figure supplement 1D*). Training did result in a decrease of behavioral and neural conflict effects in content discrimination task I of experiment 1, indicating that our training procedure was successful (*Figure 2—figure supplement 1A–C*) and suggesting more efficient functioning of conflict resolution mechanisms. In experiment 2, participants performed the volume oddball task twice, once before and once after sound location and content had been mapped to responses. Again, we aimed to see if training on conflict tasks would enhance automaticity of conflict processing in a paradigm where the auditory conflicting features were task-irrelevant. We did not find any statistically reliable differences in behavioral conflict effects or accuracy of congruency decoding between the two runs (*Figure 4—figure supplement 1*). Therefore, it seems that the automaticity of conflict detection by the MFC and associated networks does not hold when the auditory stimulus is task-irrelevant (at least after the extent of training and exposure as studied here).

## Increased detection performance on conflicting trials

Remarkably, we report increased behavioral performance on the volume oddball task for incongruent trials as compared to congruent trials (*Figure 4A*, *Figure 4—figure supplement 1B–D*). Speculatively, this increased behavioral performance (d') on incongruent trials may be due to attentional capture of conflicting stimuli. Attentional capture is the involuntary shift of attention towards salient stimuli (*Awh et al., 2012*; *Theeuwes, 2010*). Possibly, the detection of conflict between sound content and location is a salient event causing (re-)capture of attention towards the auditory stimulus, resulting in better processing of the stimulus information and ultimately better oddball detection performance. Thus, following the detection of conflict, frontal networks would have to exert control over attentional resources and direct them towards the source of the conflict. Interestingly, cases of frontal control over attentional processes have been demonstrated in the past, for example, showing that task-irrelevant distractors that have been related to reward induce stronger attentional capture (*Anderson et al., 2011*) and that high working memory load increases the strength of attentional capture by distractors (*Lavie and Fockert, 2006*). The present study was however not optimized to test directly which underlying mechanisms are associated with increased sensitivity of conflicting sensory input, and this issue merits further experimentation.

## Conclusion

Summarizing, high-level cognitive processes that require the integration of conflict inducing stimulus features are strongly hampered when none of the stimulus features of the conflict inducing stimulus are task-relevant and hence object-based attention is absent. This work nicely extends previous findings of perceptual processing outside the scope of attention (*Peelen et al., 2009*; *Sand and Wiens, 2011*; *Schnuerch et al., 2016*; *Tusche et al., 2013*), but suggests crucial limitations of the brain's capacity to process task-irrelevant 'complex' cognitive control-initiating stimuli, indicative of an attentional bottleneck to detect conflict at high levels of information analysis. In contrast, the processing of more basic physical features of sensory input appears to be less deteriorated when input is task-irrelevant (*Lachter et al., 2004*).

# Materials and methods

## Participants

We performed two separate experiments, each containing multiple behavioral tasks. For each of these experiments, we recruited 24 healthy human participants from the University of Amsterdam. None of the participants took part in both experiments. So, in total 48 participants (37 females) aged 18–30 participated in this experiment for monetary compensation or participant credits. All participants had normal or corrected-to-normal vision and had no history of head injury or physical and mental illness. This study was approved by the local ethics committee of the University of Amsterdam, and written informed consent was obtained from all participants after explanation of the experimental protocol. We will describe the experimental design and procedures for the two experiments separately. Data analyses and statistics were similar across experiments and will be discussed in the same section.

## Experiment 1: design and procedures

Participants performed two tasks in which conflicting auditory stimuli were either task-relevant or task-irrelevant. In both tasks, conflict was elicited through a paradigm adapted from *Buzzell et al., 2013*, in which spatial information and content of auditory stimuli could interfere. In content discrimination task I, participants had to respond to the auditory stimuli, whereas in vertical RDM task participants had to perform a demanding RDM task, while the auditory conflicting stimuli were still presented (*Figure 1A*). Participants performed both tasks on two experimental sessions of approximately 2.5 hr. In between these two experimental sessions, participants had two training sessions of 1 hr during which they only performed the task-relevant task (*Figure 1B*). On each experimental session, participants first performed a shortened version of the RDM task to determine the appropriate coherency of the moving dots (73–77% correct), followed by the actual task-irrelevant auditory task, and finally the task-relevant auditory task. Participants were seated in a darkened, sound-isolated room, 50 cm from a 69 × 39 cm screen (frequency: 120 Hz, resolution: 1920 × 1080, RGB: 128, 128, 128). Both tasks were programmed in MATLAB (R2012b, The MathWorks, Inc), using functionalities from Psychtoolbox (*Kleiner et al., 2007*).

## Experiment 1: behavioral tasks

### Auditory content discrimination task I

In the task-relevant auditory conflict task, the spoken words 'links' (i.e., 'left' in Dutch) and 'rechts' (i. e., 'right' in Dutch) were presented through speakers located on both sides of the participant (*Figure 1A*). Auditory stimuli were matched in duration and sample rate (44 kHz) and were recorded by the same male voice. By presenting these stimuli through either the left or the right speaker, content-location conflict arose on specific trials (e.g., the word 'left' through the right speaker). Trials were classified accordingly as either congruent (i.e., location and content are the same) or incongruent (i.e., location and content are different). Participants were instructed to respond as fast and accurate as possible by pressing left ('a') or right ('l') on a keyboard located in front of the participants, according to the stimulus content, ignoring stimulus location. Responses had to be made with the left or right index finger, respectively. The task was divided into 12 blocks of 100 trials each, allowing participants to rest in between blocks. After stimulus presentation, participants had a 2 s period in which they could respond. A variable inter-trial interval between 850 and 1250 ms was initiated directly after the response. If no response was made, the subsequent trial would start after the 2 s response period. Congruent and incongruent trials occurred equally often (i.e., 50% of all trials) as expectancy of conflict has been shown to affect conflict processing (*Soutschek et al., 2015*). Due to an error in the script, there was a disbalance in the amount of trials coming from the left (70%) versus right (30%) speaker location for the first 14 participants. However, the amount of congruent versus incongruent and 'left' versus 'right' trials was equally distributed. For the upcoming analyses, all trial classes were balanced in trial count.

## Vertical random-dot-motion task

In the task-irrelevant auditory task, participants performed an RDM task in which they had to discriminate the motion (up vs. down) of white dots (n = 603) presented on a black circle (RGB: 0, 0, 0; ~14°

visual angle; *Figure 1A*). Onset of the visual stimulus was paired with the presentation of the auditory conflicting stimulus. Participants were instructed to respond according to the direction of the dots by pressing the 'up' or 'down' key on a keyboard with their right hand as fast and accurate as possible. Again, participants could respond in a 2 s time interval, which was terminated after responses, and followed by an inter-trial interval of 850–1250 ms. Task difficulty, in terms of dot-motion coherency (i.e., proportion of dots moving in the same direction), was titrated between blocks to 73–77% correct of all trials within that block. Similar to content discrimination task I, the vertical RDM was divided into 12 blocks containing 100 trials each, separated by short breaks. Again, congruent and incongruent trials, with respect to the auditory stimuli, occurred equally often.

## Experiment 2: design and procedures

In the second experiment, we wanted to investigate whether it is task irrelevance of the auditory stimulus itself or task irrelevance of the auditory features (i.e., content and location) that determine whether prefrontal control processes are hampered. Participants performed two tasks in which auditory stimuli were fully task-relevant (location discrimination and content discrimination), one task in which the auditory stimulus was relevant but the features auditory location and content were not (volume oddball) and one task in which the auditory stimulus itself was task-irrelevant but its features location and content could potentially interfere with behavior (horizontal RDM). Participants came to the lab for one session, lasting 3 hr. Each session started with the volume oddball task, followed by (in a counterbalanced order) the location discrimination, content discrimination, and horizontal RDM tasks, and ended with a block of the volume oddball task again.

We included the location discrimination and content discrimination tasks both to replicate the results of experiment 1 and also to see if there would be differences in these results between the two tasks. Specifically, we investigated whether processing of auditory stimulus features – as indicated by multivariate classification accuracies – would differ between the two tasks. Participants were seated in a darkened, sound-isolated room, 50 cm from a 69 × 39 cm screen (frequency: 120 Hz, resolution: 1920 × 1080, RGB: 128, 128, 128). All four experiments were programmed in Python 2.7 using functionalities from PsychoPy (*Peirce et al., 2019*).

## Experiment 2: behavioral tasks

### Auditory content discrimination task (II)

The auditory content discrimination task from experiment 2 is a (near identical) replication of the auditory content discrimination task from experiment 1. Participants were fixating on a fixation mark in the center of the screen. Again, the spoken words 'links' (i.e., 'left' in Dutch) and 'rechts' (i.e., 'right' in Dutch) were presented through speakers located on both sides of the participant. Participants were instructed to respond according to the stimulus content by pressing left ('A') or right ('L') on a keyboard located in front of them, with their left and right index fingers, respectively. Concurrently, on every trial, a black disk with randomly moving dots (coherence: 0) was presented to keep sensory input similar between tasks. After stimulus presentation, participants had an 800 ms period in which they could respond. After a response, the response window would be terminated directly. A variable inter-trial interval (ITI) between 250 and 450 ms was initiated directly after the response. If no response was made, the subsequent trial would start after the ITI. All stimulus features (i.e., sound content, location, and congruency) were presented in a balanced manner (e.g., 50% congruent, 50% incongruent). The task was divided into six blocks of 100 trials each, allowing participants to rest in between blocks.

### Auditory location discrimination task

The auditory location discrimination task was identical to the auditory content discrimination task II, with the exception that participants were now instructed to respond according to the location of the auditory stimulus. Thus, participants had to press a left button ('A') for sounds coming from a left speaker and right button ('L') for sounds coming from a right speaker. Again, participants performed six blocks of 100 trials.

## Volume oddball task

In the volume oddball task, the same auditory stimuli were presented. Again, on every trial, a black disk with randomly moving dots (coherence: 0) was presented to keep sensory input similar between tasks. Occasionally an auditory stimulus would be presented at a lower volume. The initial volume of the oddballs was set to 70%, but was staircased in between blocks to yield 83–87% correct answers. If participants' performance on the previous block was below or above this range, the volume increased or decreased with 5%, respectively. The odds of a trial being an oddball trial were 1/8 (drawn from a uniform distribution). Participants were instructed to detect these oddballs by pressing the spacebar as fast as possible whenever they thought they heard a volume oddball. If they thought that the stimulus was presented at a normal volume, they were instructed to refrain from responding. The response interval was 800 ms, which was terminated at response. A variable inter-trial interval of 150–350 ms was initiated after this response interval. Participants performed two runs of this task, at the beginning of each session and at the end of each session. Each run contained five blocks of 100 trials each.

## Horizontal RDM task

In the horizontal RDM task, participants had to discriminate the motion (left vs. right) of white dots (n = 603) presented on a black circle (RGB: 0, 0, 0; ~14˚ visual angle). Onset of the visual stimulus was paired with the presentation of the auditory stimulus. Participants were instructed to respond according to the direction of the dots by pressing a left key ('A') or right key ('L') on a keyboard with left and right index fingers, respectively, as fast and accurate as possible. Participants could respond in an 800 ms time interval, which was terminated after responses, and followed by an inter-trial interval of 250–450 ms. Task difficulty, in terms of dot-motion coherency (i.e., proportion of dots moving in the same direction), was set to 0.3 in the first block. This value indicated an intermediate coherence as every trial in that block could be the intermediate coherence, but also half (0.15) or twice this intermediate coherence (0.6). The intermediate coherence was titrated between blocks to 73–77% correct. If behavioral performance fell outside that range, 0.01 was added to or subtracted from the intermediate coherence. The horizontal RDM consisted of 10 blocks of 60 trials. All stimulus features (i.e., sound content, location, and congruency) were presented in a balanced manner (e.g., 50% congruent, 50% incongruent).

## Data analysis

We were primarily interested in the effects of congruency of the auditory stimuli on both behavioral and neural data. Therefore, we defined trial congruency on the basis of these auditory stimuli in all behavioral tasks of the two experiments. All behavioral analyses were programmed in MATLAB (R2017b, The MathWorks, Inc).

## Statistical analysis of behavioral data

In all tasks, trials with an RT <100 ms or >1500 ms were excluded from behavioral analyses. Missed trials were excluded in all tasks (except the volume oddball task) as well. In order to investigate whether current trial conflict effects were present under varying levels of task relevance and to inspect if training on/exposure to conflict-inducing tasks modulated such conflict effects, we performed rm-ANOVAs on different behavioral measures. For all tasks, excluding the volume oddball task, the rm-ANOVAs were performed on the ER over all trials and RTs on correct trials. For the volume oddball task, perceptual sensitivity (d'; *Green and Swets, 1966*) and RTs of correct trials (i.e., 'hit' trials) were analyzed with rm-ANOVAs. If the assumption of sphericity was violated, we applied a Greenhouse–Geisser correction. For the tasks from experiment 1, we performed these ANOVAs with task relevance, training (before vs. after) and current trial congruency as factors (2 × 2 × 2 factorial design). Additional post-hoc rm-ANOVAs, for content discrimination task I and vertical RDM task separately (2 × 2 factorial design), were used to inspect the origin of significant factors that were modulated by task relevance.

For content discrimination task I, the location discrimination task, and horizontal RDM task, we performed a rm-ANOVA with task and congruency as factors (3 × 2 factorial design). For the volume oddball task, we performed a rm-ANOVA with congruency and run number as factors (2 × 2 factorial design) on RTs, d' scores, hit rates, and false alarm rates. We also applied paired sample t-tests

comparing the difference in these variables between incongruent and congruent for all tasks, within each experimental session (vertical RDM, content discrimination task I) and run (volume oddball task).

To test interference of the *individual* auditory features, sound content and sound location, on performance on the vertical RDM task and volume oddball task, we performed rm-ANOVAs on RTs with sound location and sound content as factors (2 × 2 factorial design). Additionally, for the horizontal RDM, we tested for auditory-visual conflict effects (i.e., conflict between sound content/location and dot direction) in RT and ER with paired sample t-tests comparing incongruent and congruent trials.

In case of null findings, we performed a Bayesian analysis (rm-ANOVA or paired sample t-test) with identical parameters and settings on the same data to test if there was actual support of the null hypothesis (*JASP Team, 2018*).

## Analysis of EEG data

EEG data were analyzed using custom-made software written in MATLAB, with support from the toolboxes EEGLAB (*Delorme and Makeig, 2004*) and ADAM (*Fahrenfort et al., 2018*).

## Acquisition and preprocessing

EEG data were recorded with a 64-channel BioSemi apparatus (BioSemi B.V., Amsterdam, The Netherlands) at 512 Hz. Vertical eye movements were recorded with electrodes located above and below the left eye, and horizontal eye movements were recorded with electrodes located at the outer canthi of the left and the right eye. All EEG traces were re-referenced to the average of two electrodes located on the left and right earlobes (mastoidal reference for one participant in experiment 1). The data were band-pass filtered offline, with cutoff frequencies of 0.01–50 Hz. Next, epochs were created by taking data from −1 s to 2 s around onset of stimulus presentation. We then rejected epochs containing blink artifacts and high-voltage artifacts. Blinks were defined as VEOG data exceeding a threshold of ±100 mV in a time window of 0–800 ms post-stimulus. This procedure resulted in the removal of 5.03% (SD = 9.71%) of epochs in experiment 1% and 6.10% (*SD* = 7.54%) of epochs in experiment 2. Subsequently, high-voltage artifacts were defined as events where voltage exceeded a threshold of ±300 mV in a time window of 0–800 ms post-stimulus on any EEG channel. With this second round of artifact rejection, 3.65% (*SD* = 7.96%) of all epochs were removed in experiment 1% and 4.16% (*SD* = 8.86%) in experiment 2. In total, 8.68% (*SD* = 12.09%) of all trials were removed in experiment 1% and 10.26% (*SD* = 13.92%) were removed in experiment 2. Note that this procedure ensures the absence of blinks and high-amplitude artifacts within the predefined ROI time window of 100–700 ms. The data of one participant in experiment 2 contained many artifacts. Specifically, across all five tasks performed in experiment 2, 64.46% (*SD* = 9.88%) of all epochs were rejected for this participant. This was more than three standard deviations from the average number of rejected trials across participants and files and left too few trials for the decoding analysis. Therefore, this participant was removed from all EEG analyses.

## Time-frequency-domain multivariate pattern analysis (decoding)

We applied a multivariate backwards decoding model to EEG data that was transformed to the time-frequency domain. We used multivariate analyses both because its higher sensitivity in comparison with univariate analyses and to inspect if and to what extent different stimulus features (i.e., location and content) were processed in both tasks, without having to preselect spatial or time-frequency ROIs. The ADAM toolbox was used for time-frequency decomposition and decoding (*Fahrenfort et al., 2018*). Single-trial power spectra were computed by convolving the EEG data with a complex wavelet (wavelet size of 0.5 s) after the application of a Hann taper (epochs: −100 ms to 1000 ms, 2–30 Hz in linear steps of 2 Hz). Raw time-frequency data contained both induced and evoked power. Trials were classified according to current trial stimulus features (i.e., location and content), resulting in four trial types. As decoding algorithms are known to be time-consuming, epochs were resampled to 64 Hz. Then, we applied a decoding algorithm to the data according to a 20-fold cross-validation scheme using either stimulus location, stimulus content, or congruency as stimulus class. Specifically, a linear discriminant analysis (LDA) was trained to discriminate between stimulus classes (e.g., left vs. right speaker location, etc.). Classification accuracy was computed as the AUC, a measure derived from Signal Detection Theory (*Green and Swets, 1966*).

The multivariate classifiers were on different subsets of trials, depending on the behavioral task. For the auditory tasks (content discrimination I and II, location discrimination, and volume oddball detection), only correct trials were included in the analysis as errors tend to elicit a similar, albeit not identical, neural response as cognitive conflict and errors are more likely on incongruent trials (*Cohen and van Gaal, 2014*). For the volume oddball detection task, we additionally excluded all oddball trials, thus only testing correct rejections, in order to prevent conflict arising between responses made exclusively with the right hand and sound content and location. For the two visual tasks (horizontal and vertical RDM), we trained the classifier on all trials.

## Topographical maps of ROI decoding

Topographical maps were created in order to investigate the spatial sources of activity related to the processing of the auditory features (content, location, and congruency). We first extracted classifier weights for each task and feature from the predefined ROI (100–700 ms, 2–8 Hz), allowing us to directly compare the spatial distributions between features and tasks. However, raw classifier weights are not interpretable as neural sources of activity and therefore have to be reconstructed (*Haufe et al., 2014*). Thus, classifier weights were transformed to activation patterns by multiplying them with the covariance in the EEG data. The topographical activity maps of tasks and features with low decoding performance should be interpreted with caution as activation patterns reconstructed from classifier weights may be unreliable when decoding performance is low (*Haufe et al., 2014*).

## Time-domain multivariate pattern analysis (decoding)

We applied a time-domain decoding analysis on EEG data to inspect the possible effect of task relevance of a stimulus feature on sensory processing. For this analysis, only EEG data from the tasks of experiment 2 were used as its parameters were comparable between tasks in this experiment (e.g., always visual stimulus present, no extensive training). For the analysis, we trained linear classifiers (LDA) to discriminate sound content ('left' vs. 'right') or sound location (left speaker vs. right speaker). First, epochs (from −100 ms to 1000 ms around stimulus presentation) were resampled to 64 Hz similar to the time-frequency decoding analyses. Then, the models were trained and tested according to a 20-fold cross-validation scheme. The AUC scores we obtained via multivariate analyses of our EEG data were tested per timepoint with one-sided t-tests ($\bar{X}>0.5$) across participants against chance level (50%). These t-tests were corrected for multiple comparisons over time using cluster-based permutation tests (p<0.05, 1000 iterations). For each decoded stimulus feature, we then compared the decoding accuracies of the behavioral task in which the feature was task-relevant to all other tasks in a pairwise fashion (e.g., location decoding under location discrimination task vs. horizontal RDM task), with cluster-corrected two-sided t-tests against 0.

## Statistical analysis of EEG data

The AUC scores we obtained via multivariate analyses of our EEG data were tested per timepoint and frequency with one-sided t-tests ($\bar{X}>0.5$) across participants against chance level (50%). These t-tests were corrected for multiple comparisons over time and frequency using cluster-based permutation tests (p<0.05, 1000 iterations). This procedure yields time-frequency clusters of significant above-chance classifier accuracy, indicative of information processing. Note that this procedure yields results that should be interpreted as fixed effects (*Allefeld et al., 2016*), but is nonetheless standard in the scientific community.

In addition to the cluster analysis, we performed hypothesis-driven analyses on classifier accuracies that were extracted from a predefined time-frequency ROI. All these analyses were performed in JASP (*JASP Team, 2018*). We then applied an ANCOVA on these accuracies with fixed effects being task and stimulus feature. Next, one-sample t-tests (one-sided, $\bar{X}>0.5$) were performed on every task/feature combination to determine whether decoding accuracy of a specific feature within our preselected ROI was above chance during the various behavioral tasks. Additional Bayesian one-sample t-tests (one-sided, $\bar{X}>0.5$, Cauchy scale = 0.71) were performed to inspect evidence in favor of the null hypothesis that decoding accuracy was not above chance.

We performed the same analysis on different ROIs. The results of those analyses can be found in *Figure 5—figure supplement 1B–D*.

## Additional information

### Funding

| Funder | Grant reference number | Author |
|---|---|---|
| H2020 European Research Council | ERC-2016-STG_715605 | Simon van Gaal |

The funders had no role in study design, data collection and interpretation, or the decision to submit the work for publication.

### Author contributions
Stijn Adriaan Nuiten, Conceptualization, Formal analysis, Investigation, Visualization, Writing - original draft; Andrés Canales-Johnson, Tristan Bekinschtein, Conceptualization, Writing - review and editing; Lola Beerendonk, Data curation, Writing - review and editing; Nutsa Nanuashvili, Investigation; Johannes Jacobus Fahrenfort, Conceptualization, Formal analysis, Writing - review and editing; Simon van Gaal, Conceptualization, Resources, Supervision, Funding acquisition, Writing - review and editing

### Author ORCIDs
Stijn Adriaan Nuiten (iD) https://orcid.org/0000-0002-9248-166X
Andrés Canales-Johnson (iD) https://orcid.org/0000-0002-2747-8894
Lola Beerendonk (iD) https://orcid.org/0000-0002-4095-5122
Nutsa Nanuashvili (iD) https://orcid.org/0000-0003-2408-1821
Johannes Jacobus Fahrenfort (iD) http://orcid.org/0000-0002-9025-3436
Simon van Gaal (iD) https://orcid.org/0000-0001-6628-4534

### Ethics
Human subjects: Written informed consent was obtained from all participants after explanation of the experimental protocol. This study was approved by the local ethics committee of the University of Amsterdam (projects: 2015-BC-4687, 2017-BC-8257, 2019-BC-10711).

### Decision letter and Author response
Decision letter https://doi.org/10.7554/eLife.64431.sa1
Author response https://doi.org/10.7554/eLife.64431.sa2

## Additional files
### Supplementary files
• Transparent reporting form

### Data availability
The data and analysis scripts used in this article is available on Figshare: https://uvaauas.figshare.com/projects/Preserved_sensory_processing_but_hampered_conflict_detection_when_stimulus_input_is_task-irrelevant/115020.

The following datasets were generated:

| Author(s) | Year | Dataset title | Dataset URL | Database and Identifier |
|---|---|---|---|---|
| Nuiten SA, Canales-Johnson A, Beerendonk L, Nanuashvili N, Fahrenfort JJ, Bekinschtein T, van Gaal S | 2021 | Analyses scripts for manuscript: Preserved sensory processing but hampered conflict detection when stimulus input is task-irrelevant | https://doi.org/10.21942/uva.14730297 | figshare, 10.21942/uva.14730297 |

| Nuiten SA, Canales-Johnson A, Beerendonk L, Nanuashvili N, Fahrenfort JJ, Bekinschtein T, van Gaal S | 2021 | Raw behavioral dataset for manuscript: Preserved sensory processing but hampered conflict detection when stimulus input is task-irrelevant | https://www.narcis.nl/dataset/RecordID/doi%3A10.21942%2Fuva.14709420.v1 | DANS, 14709420.v1 |
| Nuiten SA, Canales-Johnson A, Beerendonk L, Nanuashvili N, Fahrenfort JJ, Bekinschtein T, van Gaal S | 2021 | Decoded EEG (time-frequency) dataset for manuscript: Preserved sensory processing but hampered conflict detection when stimulus input is task-irrelevant | https://doi.org/10.21942/uva.14730402.v1 | figshare, 10.21942/uva.14730402.v1 |
| Nuiten SA, Canales-Johnson A, Beerendonk L, Nanuashvili N, Fahrenfort JJ, Bekinschtein T, van Gaal S | 2021 | Decoded EEG (time-domain) dataset for manuscript: Preserved sensory processing but hampered conflict detection when stimulus input is task-irrelevant | https://doi.org/10.21942/uva.14754870.v1 | figshare, 10.21942/uva.14754870.v1 |
| Nuiten SA, Canales-Johnson A, Beerendonk L, Nanuashvili N, Fahrenfort JJ, Bekinschtein T, van Gaal S | 2021 | Raw EEG dataset for manuscript: Preserved sensory processing but hampered conflict detection when stimulus input is task-irrelevant | https://doi.org/10.21942/uva.14709420.v1 | figshare, 10.21942/uva.14709420.v1 |

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
