## [Decision Letter]

**Acceptance summary:**

Nuiten and colleagues have conducted a well-designed series of electroencephalographic experiments to investigate if conflict detection depends on conscious awareness. They used a combination of behavioral findings and sophisticated modeling of EEG data to conclude that conflict was only present when there was at least some degree of task relevance and that there was a dependence on object-based attention. In contrast, sensory processing was preserved regardless of attentional status.

**Decision letter after peer review:**

Thank you for submitting your article "Intact sensory processing but hampered conflict detection when stimulus input is task-irrelevant" for consideration by *eLife*. Your article has been reviewed by 3 peer reviewers, and the evaluation has been overseen by a Reviewing Editor and Michael Frank as the Senior Editor. The following individuals involved in review of your submission have agreed to reveal their identity: Ulrike M. Krämer, PhD (Reviewer #1); Jason Samaha (Reviewer #2).

The reviewers have discussed the reviews with one another and the Reviewing Editor has drafted this decision to help you prepare a revised submission.

Summary:

Nuiten and colleagues have conducted a well-designed series of EEG experiments to investigate if conflict detection depends on conscious awareness. They used a combination of behavioral findings and MVPA analysis of EEG data to conclude that conflict was only present when there was at least some degree of task relevance and that there was a dependence on object-based attention. In contrast, sensory processing was intact regardless of attentional status.

Overall, the reviewers were enthusiastic about the manuscript. However, there were some concerns and questions which we discuss below.

Essential revisions:

1) The reviewers felt that describing the sensory processing as "intact" regardless of attention was too vague. While there does seem to be some degree of classification possible for sensory processing regardless of task relevance – it appears to be graded. For instance, there are clear differences between the decodability of auditory stimulus properties when the auditory stimulus is task-relevant versus not. Some of this can be attributed to decision/motor processes, as the authors point out, but some changes might reflect changes in sensory processing – for example, the earlier aspects of the decoding differences. In this case, the conclusion would be that sensory processes are 'relatively' intact, but potentially modulated by task-relevance. We recommend the authors show sensory ERPs when the auditory stimulus is task-relevant or not to help assess whether the task manipulation keeps sensory responses completely intact or the authors otherwise justify how their results support the idea that sensory processing is "intact".

2) The reviewers had some questions about what might be driving the congruency decoding results. Given that the congruent and non-congruent conditions produce large differences in behavior in almost all tasks (except the RDM tasks, where no conflict decoding was found), is it possible that the decoder is just picking up on the task difficulty, rather than conflict detection per se? For example, perhaps the decoder results reflect differences in decision making as a result of conflict, rather than the neural signature of the conflict detection process. Is there anything about the temporal/spatial dynamics of theta amplitude that would indicate a specific conflict detection process is underlying decodability?

3) The reviewers were unclear about the interpretation of the decoding results in terms of object-based attention. Specifically, on page 13 the authors write: "Second, the time-frequency (T-F) windows of significant sound content and location decoding in the volume oddball task was considerably more extended in both time and frequency space than observed in the two RDM tasks. This highlights that the presence of object-based attention in the volume oddball task, because one feature of the auditory stimulus was task-relevant, led to the rapid attentional selection and hence neural enhancement of the task-irrelevant features sound content and location this task only". If this explanation were true, and object-based attention was the mechanism, wouldn't a similarly broad time/frequency decoding effect be observed in the content decoding during location discrimination task, since the sound is the object of attention in that task as well? However, the data (4b) seem to only show a small T/F window of content decoding on the location task, suggesting that object-based decoding is either not operating in that task or that the broad T/F decoding does not actually reflect object-based selection. Can the authors comment on this apparent discrepancy?

4) The reviewers found it unfortunate that information about which electrodes contributed to decoding was lost in this analysis and felt that the manuscript would be improved if at least some topographical information were included.

5) The authors state that results in Figure 5 are based on a specific time-frequency ROI which was hypothesis-driven and pre-defined (p.24, lines 911). However, they go on in the next sentence to explain that the ROI was selected visually based on the most-significant clusters. Please clarify how the ROI was defined – was it hypothesis-driven or data-driven? If it was data-driven how were multiple comparison corrections handled?

6) The reviewers were surprised by the choice to not perform any artifact rejection on the data. We strongly suggest that appropriate artifact reject be applied, or, at the very least, the choice to not perform artifact rejection be better justified.

---

## [Author Response]

Essential revisions:1) The reviewers felt that describing the sensory processing as “intact” regardless of attention was too vague. While there does seem to be some degree of classification possible for sensory processing regardless of task relevance – it appears to be graded. For instance, there are clear differences between the decodability of auditory stimulus properties when the auditory stimulus is task-relevant versus not. Some of this can be attributed to decision/motor processes, as the authors point out, but some changes might reflect changes in sensory processing – for example, the earlier aspects of the decoding differences. In this case, the conclusion would be that sensory processes are ‘relatively’ intact, but potentially modulated by task-relevance. We recommend the authors show sensory ERPs when the auditory stimulus is task-relevant or not to help assess whether the task manipulation keeps sensory responses completely intact or the authors otherwise justify how their results support the idea that sensory processing is “intact”.

We agree with the reviewers that, on the basis of the results presented in the manuscript, we cannot claim “intact” sensory processing regardless of task-relevance. The use of the word “*intact”* may indeed suggest that the processing of sensory features is unchanged under various levels of task-relevance, which we did not mean to bring across. Indeed, a substantial body of evidence has shown reduced evoked responses to stimuli that were not attended or were task-irrelevant (Alilović et al., 2019; Jehee et al., 2011; Kok et al., 2012; Molloy et al., 2015b; Woldorff et al., 1993). Our time-frequency decoding results point to the same idea, e.g. location decoding is more broadband and better when the auditory stimulus is task-relevant versus when it is not (e.g. manuscript Figure 2B). We understand the reviewers’ suggestion to inspect sensory ERPs, to test this issue further. As such, we have now performed an additional analysis to inspect the early sensory evoked responses as a factor of task-relevance. We decided to use a time-domain decoding approach however, instead of a more traditional ERP analysis, for two reasons. We wanted to prevent obscuring possible sensory effects due to electrode selection, necessary for computing ERPs but not for decoding, because we did not have strong a priori expectations about the exact scalp topography of the processing of sound content and location as a factor of task-relevance. Second, because we have used multivariate approaches throughout the manuscript so far, this better fitted the general approach taken in this project. Below, we have added a paragraph of the Methods section where we describe the details of the time-domain decoding analysis. Also, we added excerpts of the manuscript (Results) where we present the results of this new analysis (Figure 4 – Supplement 2).

As you will see, these additional analyses suggest that sensory processing is indeed graded, although it remains difficult to know with certainty, because of the direct association between specific responses/decision processes and specific features in some tasks but not others. We thus agree with the reviewers that the effects we report in the manuscript (manuscript Figures 2, 4 and 5; Figure 4—figure supplement 2) need more subtle phrasing. We have decided to refer to sensory processing as *preserved* instead of *intact* throughout the manuscript. As such, we have changed the title of the revised manuscript into ‘Preserved sensory processing but hampered conflict detection when stimulus input is task-irrelevant’. We hope this better qualifies the graded relationship between sensory processing and task-relevance. Also, we have added a paragraph to the Discussion where we discuss the graded sensory processing as a factor of task-relevance and what that means for our reported findings. We have also added that paragraph below.

In Methods section, page 30

“We applied a time-domain decoding analysis on EEG data, to inspect the possible effect of task relevance of a stimulus feature on sensory processing. […] For each decoded stimulus feature, we then compared the decoding accuracies of the behavioral task in which the feature was task-relevant, to all other tasks in a pairwise fashion (e.g. location decoding under location discrimination task versus horizontal RDM task), with cluster-corrected two-sided t-tests against 0.”

In Results section, page 17

“In order to test whether the earliest sensory responses were already modulated by task relevance, and to link this to previous ERP studies (Alilović et al., 2019; Molloy et al., 2015; Woldorff et al., 1993), we performed an additional time-domain multivariate analysis on these sensory features (T-F analysis are not well suited to address questions about the timing of processes). […] These results are elaborated upon in the Discussion.”

In Discussion, pages 21-22

“We show that conflict processing is absent when conflicting features are fully task-irrelevant, while evidence of sensory processing is still present in neural data (Figures 2B, 4B and 5). […] Summarizing, although processing of sensory features is degraded under decreasing levels of task relevance it is present regardless of attention, whereas detection of conflict between these features is no longer possible when the features are fully task-irrelevant.”

2) The reviewers had some questions about what might be driving the congruency decoding results. Given that the congruent and non-congruent conditions produce large differences in behavior in almost all tasks (except the RDM tasks, where no conflict decoding was found), is it possible that the decoder is just picking up on the task difficulty, rather than conflict detection per se? For example, perhaps the decoder results reflect differences in decision making as a result of conflict, rather than the neural signature of the conflict detection process. Is there anything about the temporal/spatial dynamics of theta amplitude that would indicate a specific conflict detection process is underlying decodability?

We thank the reviewers for this interesting question. The reviewers point to an issue concerning the source of our above-chance decoding of congruency. Whether typical neural effects in research on conflict processing, i.e. enhanced medial frontal theta-oscillations, are related to the detection of conflict or are a marker of task-difficulty has been debated in the field (Grinband et al., 2011a, 2011b; Yeung et al., 2011). We agree that this distinction is important and deserves more elaboration in the manuscript. We have therefore added a paragraph to the Discussion in which we address this issue. We have added that paragraph below.

In Discussion, page 22

“For our main analysis we trained a multivariate classifier on congruent versus incongruent trials and observed effects of task relevance of the performance of the classifier, i.e. decoding performance was hampered when conflicting features were fully task-irrelevant (Figures 2B, 4B and 5B). […] Therefore, we believe that the observed congruency decoding results presented here are mainly driven by the detection of conflicting sensory inputs and are not, or much less so, driven by task difficulty.”

3) The reviewers were unclear about the interpretation of the decoding results in terms of object-based attention. Specifically, on page 13 the authors write: "Second, the time-frequency (T-F) windows of significant sound content and location decoding in the volume oddball task was considerably more extended in both time and frequency space than observed in the two RDM tasks. This highlights that the presence of object-based attention in the volume oddball task, because one feature of the auditory stimulus was task-relevant, led to the rapid attentional selection and hence neural enhancement of the task-irrelevant features sound content and location this task only". If this explanation were true, and object-based attention was the mechanism, wouldn't a similarly broad time/frequency decoding effect be observed in the content decoding during location discrimination task, since the sound is the object of attention in that task as well? However, the data (4b) seem to only show a small T/F window of content decoding on the location task, suggesting that object-based decoding is either not operating in that task or that the broad T/F decoding does not actually reflect object-based selection. Can the authors comment on this apparent discrepancy?

This is a good point, and we thank the reviewers for this comment. After reconsideration of our results, we agree with the reviewers that object-based attention alone cannot explain why sensory feature decoding is more durable and broadband when the auditory stimulus is task-relevant versus task-irrelevant, as it falls short in explaining the more fleeting and narrowband decoding results for task-irrelevant auditory features (e.g. content) during the auditory tasks (e.g. location discrimination task). We have therefore removed this explanation of the decoding results (previous lines 448-453) and supplementary figure S5 from the manuscript. Moreover, we have added a paragraph to the Discussion in which we discuss a possible second mechanism, besides object-based attention, that may be at play and may affect the decoding results, depending on the specific task manipulation. We have added that paragraph below.

In Discussion, page 22

“Besides object-based attention, the process through which attentional resources are allocated to the processing of task-irrelevant features of a task-relevant object, other mechanisms might also play a role in the extent to which sensory information is processed, such as the active suppression of task-irrelevant information. […] Disentangling the effects of such mechanisms, object-based attention and their possible interactions on the processing of sensory and cognitive information, however, falls outside the scope of this work.”

4) The reviewers found it unfortunate that information about which electrodes contributed to decoding was lost in this analysis and felt that the manuscript would be improved if at least some topographical information were included.

We agree with the reviewers that information pertaining to the spatial sources of our effects could provide additional insights. The decoding weights of EEG-channels are, however, not interpretable as neural sources and therefore they have to be transformed back to activity patterns (Haufe et al., 2014). In order to compare topographic maps for all tasks and features, we have extracted the decoder weights from the preselected time-frequency ROI used for manuscript figure 5 for all tasks and contrasts (sound content, sound location and conflict). Then, these weights were transformed to activity patterns by multiplying them with the covariance in the EEG data (Haufe et al., 2014). We have added the topographic maps and the parts from the Results and Methods of the manuscript where we discuss these maps (Figure 5 – Supplement 1A).

In Methods, page 30

“Topographical maps were created in order to investigate the spatial sources of activity related to the processing of the auditory features (content, location and congruency). […] The topographical activity maps of tasks and features with low decoding performance should be interpreted with caution, as activation patterns reconstructed from classifier weights may be unreliable when decoding performance is low (Haufe et al., 2014).”

In Results, page 9

“Activation patterns that were calculated from classifier weights within the predefined time-frequency theta-band ROI (2Hz-8Hz, 100ms-700ms) revealed a clear midfrontal distribution of conflict related activity (Figure 5 – Supplement 1A).”

In Results, page 18

“Classifier weights were extracted from the ROI for all tasks and features, transformed to activation patterns and plotted in topomaps, to show the patterns of activity underlying the decoding results (Figure 5 – Supplement 1A).”

5) The authors state that results in Figure 5 are based on a specific time-frequency ROI which was hypothesis-driven and pre-defined (p.24, lines 911). However, they go on in the next sentence to explain that the ROI was selected visually based on the most-significant clusters. Please clarify how the ROI was defined – was it hypothesis-driven or data-driven? If it was data-driven how were multiple comparison corrections handled?

We apologize, the wording used in the outlined section was ambiguous and we have therefore revised it. The time-frequency ROI was selected on the basis of previous work, including our own, investigating the role of medial frontal theta-oscillations in conflict processing (Cohen and Cavanagh, 2011; Cohen and van Gaal, 2014; Jiang et al., 2015; Nigbur et al., 2012). Furthermore, we would like to emphasize that we verified whether ROI selection influenced our main findings, by applying the same analyses to other time-frequency ROIs. These findings are depicted in the supplements and show a similar pattern as the ROI presented in the main text, i.e. a stronger decline in congruency decoding accuracy under manipulations of task-relevance as opposed to decoding accuracy of sensory features. We have added the revised section below.

In Results, page 7

“Then, we report results from the additional hypothesis-driven analysis, where we extracted classifier accuracies from a predefined time-frequency ROI (100ms-700ms, 2Hz-8Hz) on which we performed (Bayesian) tests (see Methods). […] Specifically, for every task and every stimulus feature (i.e. congruency, content, location), we extracted average decoding accuracies from the ROI per participant and performed analyses on these values.”

6) The reviewers were surprised by the choice to not perform any artifact rejection on the data. We strongly suggest that appropriate artifact reject be applied, or, at the very least, the choice to not perform artifact rejection be better justified.

We have followed the reviewers’ suggestion to perform further preprocessing on our data. Thus, the data discussed in our revised manuscript, and this letter, are fully updated.

We initially did not perform any artefact rejection, as in our experience multivariate analyses on large EEG datasets are robust to artefacts. For smaller datasets, such as the data belonging to Experiment 2, we believed that preserving as much data as possible would improve classification performance by increasing the training set size of the model. However, we understand the reviewers’ suggestion to perform artefact rejection. We reasoned this would indeed strengthen the results presented in the paper, because readers may wonder whether our (null-)findings were perhaps a result of noisy data due to our lack of artefact rejection. We have now performed the same multivariate analyses as in the original version of the manuscript, but this time after additional preprocessing steps

We were pleased that the reviewers suggested to do additional preprocessing for several reasons. First, the majority of the decoding results remained qualitatively unchanged, strengthening our confidence in the robustness of these results. Interestingly, however, we do find significant congruency decoding in the predefined ROI for the horizontal RDM task after applying these preprocessing steps (manuscript Figure 4B), whereas previously no effects of congruency were observed in this task. Above-chance congruency decoding in the HRDM was robust and independent of the specific definition of the ROI used (manuscript Figure 5 – Supplement 1B-D). This observation of congruency decoding within the theta-band shows that when a task-irrelevant stimulus has features that are response-relevant (i.e. overlap with the response scheme), these two features are still integrated in the brain to form conflict. In contrast, when features of a task-irrelevant stimulus are not response-relevant, i.e. in the vertical RDM task where the response is orthogonal to sound content and location, conflict between these auditory features is not detected. We have incorporated this interesting new finding in the revised manuscript, and have updated the abstract, results and Discussion section accordingly. Below, we have added the section from the Methods where we describe the preprocessing pipeline and an excerpt from the Discussion where we discuss the new results.

In Methods, page 29

“EEG-data were recorded with a 64-channel BioSemi apparatus (BioSemi B.V., Amsterdam, The Netherlands), at 512Hz. […] This was more than 3 standard deviations from the average number of rejected trials across participants and files and left too few trials for the decoding analysis. Therefore this participant was removed from all EEG analyses.”

In Discussion, pages 20-21

“In the horizontal RDM task on the other hand, the conflicting features of the task-irrelevant auditory stimulus overlapped with the overall response scheme or task-set of the participant, namely discriminating rightwards versus leftwards moving dots. […] Note that we report additional behavioral results that show clear indications of conflict when the task-relevant feature of the visual stimulus interferes directly with a *single* task-irrelevant feature of the auditory task (e.g., auditory content-dot motion conflict).”

References:

Alilović, J., Timmermans, B., Reteig, L. C., van Gaal, S., and Slagter, H. A. (2019). No Evidence that Predictions and Attention Modulate the First Feedforward Sweep of Cortical Information Processing. Cerebral Cortex, 29(5), 2261–2278. https://doi.org/10.1093/cercor/bhz038Anderson, B. A., Laurent, P. A., and Yantis, S. (2011). Value-driven attentional capture. Proceedings of the National Academy of Sciences, 108(25), 10367–10371. https://doi.org/10.1073/pnas.1104047108Appelbaum, L. G., Smith, D. V., Boehler, C. N., Chen, W. D., and Woldorff, M. G. (2011). Rapid Modulation of Sensory Processing Induced by Stimulus Conflict. Journal of Cognitive Neuroscience, 23(9), 2620–2628. https://doi.org/10.1162/jocn.2010.21575Awh, E., Belopolsky, A. V., and Theeuwes, J. (2012). Top-down versus bottom-up attentional control: A failed theoretical dichotomy. Trends in Cognitive Sciences, 16(8), 437–443. https://doi.org/10.1016/j.tics.2012.06.010Cohen, M. X., and Cavanagh, J. F. (2011). Single-Trial Regression Elucidates the Role of Prefrontal Theta Oscillations in Response Conflict. Frontiers in Psychology, 2. https://doi.org/10.3389/fpsyg.2011.00030Cohen, M. X., and van Gaal, S. (2014). Subthreshold muscle twitches dissociate oscillatory neural signatures of conflicts from errors. NeuroImage, 86, 503–513. https://doi.org/10.1016/j.neuroimage.2013.10.033Egner, T., and Hirsch, J. (2005). Cognitive control mechanisms resolve conflict through cortical amplification of task-relevant information. Nature Neuroscience, 8(12), 1784–1790. https://doi.org/10.1038/nn1594Fahrenfort, J. J., Van Driel, J., van Gaal, S., and Olivers, C. N. L. (2018). From ERPs to MVPA using the Amsterdam Decoding and Modeling toolbox (ADAM). Frontiers in Neuroscience – Brain Imaging Methods, 12(July). https://doi.org/10.3389/fnins.2018.00368Grinband, J., Savitskaya, J., Wager, T. D., Teichert, T., Ferrera, V. P., and Hirsch, J. (2011a). Conflict, error likelihood, and RT: Response to Brown and Yeung et al. NeuroImage, 57(2), 320–322. https://doi.org/10.1016/j.neuroimage.2011.04.027Grinband, J., Savitskaya, J., Wager, T. D., Teichert, T., Ferrera, V. P., and Hirsch, J. (2011b). The Dorsal Medial Frontal Cortex is Sensitive to Time on Task, Not Response Conflict or Error Likelihood. NeuroImage, 57(2), 303–311. https://doi.org/10.1016/j.neuroimage.2010.12.027Haufe, S., Meinecke, F., Görgen, K., Dähne, S., Haynes, J.-D., Blankertz, B., and Bießmann, F. (2014). On the interpretation of weight vectors of linear models in multivariate neuroimaging. NeuroImage, 87, 96–110. https://doi.org/10.1016/j.neuroimage.2013.10.067Janssens, C., De Loof, E., Boehler, C. N., Pourtois, G., and Verguts, T. (2018). Occipital α power reveals fast attentional inhibition of incongruent distractors. Psychophysiology, 55(3). https://doi.org/10.1111/psyp.13011Jehee, J. F. M., Brady, D. K., and Tong, F. (2011). Attention Improves Encoding of Task-Relevant Features in the Human Visual Cortex. Journal of Neuroscience, 31(22), 8210–8219. https://doi.org/10.1523/JNEUROSCI.6153-09.2011Jiang, J., Zhang, Q., and van Gaal, S. (2015). Conflict awareness dissociates theta-band neural dynamics of the medial frontal and lateral frontal cortex during trial-by-trial cognitive control. NeuroImage, 116, 102–111. https://doi.org/10.1016/j.neuroimage.2015.04.062Jiang, J., Zhang, Q., and Van Gaal, S. (2015). EEG neural oscillatory dynamics reveal semantic and response conflict at difference levels of conflict awareness. Scientific Reports, 5(July), 1–11. https://doi.org/10.1038/srep12008Jiang, Jun, Zhang, Q., and Van Gaal, S. (2015). EEG neural oscillatory dynamics reveal semantic and response conflict at difference levels of conflict awareness. Scientific Reports, 5(1), 12008. https://doi.org/10.1038/srep12008Kok, P., Rahnev, D., Jehee, J. F. M., Lau, H. C., and de Lange, F. P. (2012). Attention Reverses the Effect of Prediction in Silencing Sensory Signals. Cerebral Cortex, 22(9), 2197–2206. https://doi.org/10.1093/cercor/bhr310Lavie, N., and de Fockert, J. (2006). Frontal control of attentional capture in visual search. Visual Cognition, 14(4–8), 863–876. https://doi.org/10.1080/13506280500195953McKay, C. C., van den Berg, B., and Woldorff, M. G. (2017). Neural cascade of conflict processing: Not just time-on-task. Neuropsychologia, 96, 184–191. https://doi.org/10.1016/j.neuropsychologia.2016.12.022Molloy, K., Griffiths, T. D., Chait, M., and Lavie, N. (2015a). Inattentional Deafness: Visual Load Llleads to Time-Specific Suppression of Auditory Evoked Responses. Journal of Neuroscience, 35(49), 16046–16054. https://doi.org/10.1523/JNEUROSCI.2931-15.2015Molloy, K., Griffiths, T. D., Chait, M., and Lavie, N. (2015b). Inattentional Deafness: Visual Load Leads to Time-Specific Suppression of Auditory Evoked Responses. Journal of Neuroscience, 35(49), 16046–16054. https://doi.org/10.1523/JNEUROSCI.2931-15.2015Nigbur, R., Cohen, M. X., Ridderinkhof, K. R., and Stürmer, B. (2012a). Theta Dynamics Reveal Domain-specific Control over Stimulus and Response Conflict. Journal of Cognitive Neuroscience, 24(5), 1264–1274. https://doi.org/10.1162/jocn_a_00128Nigbur, R., Cohen, M. X., Ridderinkhof, K. R., and Stürmer, B. (2012b). Theta Dynamics Reveal Domain-specific Control over Stimulus and Response Conflict. Journal of Cognitive Neuroscience, 24(5), 1264–1274. https://doi.org/10.1162/jocn_a_00128Polk, T. A., Drake, R. M., Jonides, J. J., Smith, M. R., and Smith, E. E. (2008). Attention Enhances the Neural Processing of Relevant Features and Suppresses the Processing of Irrelevant Features in Humans: A Functional Magnetic Resonance Imaging Study of the Stroop Task. The Journal of Neuroscience, 28(51), 13786–13792. https://doi.org/10.1523/JNEUROSCI.1026-08.2008Ruggeri, P., Meziane, H. B., Koenig, T., and Brandner, C. (2019). A fine-grained time course investigation of brain dynamics during conflict monitoring. Scientific Reports, 9(1), 3667. https://doi.org/10.1038/s41598-019-40277-3Theeuwes, J. (2010). Top–down and bottom–up control of visual selection. Acta Psychologica, 135(2), 77–99. https://doi.org/10.1016/j.actpsy.2010.02.006Woldorff, M. G., Gallen, C. C., Hampson, S. A., Hillyard, S. A., Pantev, C., Sobel, D., and Bloom, F. E. (1993). Modulation of early sensory processing in human auditory cortex during auditory selective attention. Proceedings of the National Academy of Sciences, 90(18), 8722–8726. https://doi.org/10.1073/pnas.90.18.8722Yeung, N., Cohen, J. D., and Botvinick, M. M. (2011). Errors of interpretation and modeling: A reply to Grinband et al. NeuroImage, 57(2), 316–319. https://doi.org/10.1016/j.neuroimage.2011.04.029